

# Integrating canopy and large-scale atmospheric effects in convective boundary-layer dynamics during CHATS experiment

Metodija M. Shapkalijevski[1,2], G. Huug Ouwersloot[2], F. Arnold Moene[1], Jordi Vilà-Guerau Arellano[1,2]

[1]Meteorology and Air Quality section (MAQ), Wageningen University, Wageningen, 6700AA, The Netherlands

[2]Max-Planck Institute for Chemistry (MPIC), Mainz, 55128, Germany

*Correspondence to*: Metodija M. Shapkalijevski (meto.shapkalijevski@wur.nl)

**Abstract.** By characterizing the dynamics of a convective boundary layer above a relatively sparse and uniform orchard canopy, we investigated the impact of the roughness sublayer (RSL) representation on the predicted diurnal variability of surface fluxes and state variables. Our approach combined numerical experiments, using an atmospheric mixed-layer model

including a land surface-vegetation representation, and measurements from the Canopy Horizontal Array Turbulence Study (CHATS) field experiment near Dixon, California. The RSL is parameterized using an additional factor in the standard Monin-Obukhov Similarity Theory flux-profile relationships that takes into account the canopy's influence on the atmospheric flow. We selected a representative case characterised by southerly wind conditions to ensure well-developed RSL over the orchard canopy. We then investigated the sensitivity of the diurnal variability of the boundary-layer dynamics

to the changes in the RSL key scales, the canopy adjustment length scale, $L_c$, and the $\beta = u_*/|U|$ ratio at the top of the canopy, due to their stability and dependence on canopy structure. We found that the inclusion of the RSL parameterisation resulted in improved prediction of the diurnal evolution of the near-surface mean quantities (e.g. up to 50 % for the wind velocity) and transfer (drag) coefficients. We found relatively insignificant effects on the modelled surface fluxes (e.g. up to 5 % for the friction velocity, while 3 % for the sensible and latent heat), which is due to the compensating effect between the

mean gradients and the drag coefficients, which are both largely affected by the RSL parameterisation. When varying $L_c$ (from 10 to 20m) and $\beta$ (from 0.25 to 0.4), based on observational evidence, the predicted friction velocity is found to vary by up to 25 % and the modelled surface energy fluxes (SH and LE) vary up to 2 % and 9 %, respectively. Consequently, the boundary-layer height varies up to 6 %. Furthermore, our analysis indicated that to interpret the CHATS measurements above the canopy, the contributions of non-local effects such as entrainment, subsidence and the advection of heat and

moisture over the CHATS site need to be taken into account.

## 1 Introduction

The atmospheric boundary layer (ABL), as a part of the global climate, is a dynamic system that is highly dependent on the turbulent exchange of energy, momentum and matter between the Earth's surface and the free troposphere, as well as on the





influence of larger-scale atmospheric processes (Stull, 2009). Tall plant canopies modify turbulence at the canopy-atmosphere interface, leading to specific turbulent organised structures (Raupach et al., 1996). These coherent turbulent structures in the canopy vicinity are similar in nature to eddies developed in a plane mixing layer (Raupach et al., 1996; Finnigan, 2000; Finnigan et al., 2009). The layer in which these turbulent structures appear and affect the atmospheric flow

is called the roughness sublayer (RSL). These structures are responsible for most of the momentum (70%) and turbulent kinetic energy (90 %) exchange between canopy and atmosphere (Finnigan, 2000; Finnigan et al., 2009), and depend on canopy density as well as atmospheric diabatic stability (Dupont and Patton, 2012b; Shapkalijevski et al. 2016). Representing the ABL dynamics, considering the RSL turbulence within the system, may be of importance in numerical weather prediction models (NWP) (Physick and Garratt, 1995; Harman, 2012).

A number of observational studies have demonstrated the importance of canopy effects on the turbulent exchange of energy, mass and momentum within the RSL for different canopy types (e.g., Thom et al., 1975; Raupach, 1979; Denmead and Bradley, 1985; Högström et al., 1989). They all pointed out the failure of the traditional Monin-Obukhov similarity theory (MOST, Monin and Obukhov, 1954) to link turbulent fluxes to the mean profiles within the RSL. To account for the canopy effects, a number of different formulations parameterising the effect of RSL have been proposed to modify the standard

MOST flux-profile relationships (Garratt, 1980; Cellier and Brunet, 1992; Raupach, 1992; Mölder et al., 1999; Harman and Finnigan, 2007, 2008; De Ridder, 2010). The latter resulted in improved flux calculations just above the canopy when inferred from profile measurements (Mölder et al., 1999; De Ridder, 2010).

The flux-profile relationships are commonly used within the surface scheme of the atmospheric models. There have been efforts to incorporate the effect of RSL turbulence, by using the above-mentioned RSL-adapted flux-profile relationship, in

the surface schemes of numerical atmospheric models (Physick and Garratt, 1995; Harman, 2012). Physick and Garratt (1995), who incorporated a relatively simple RSL parameterization within the surface scheme of a mesoscale model, studied the impact of the RSL on the deposition velocity and mean variables above the canopy. Physick and Garratt (1995) found significant variation in mean wind speed within the RSL, while only small (less than 3 %) on surface fluxes. Harman (2012) later implemented a more physically sound RSL formulation (based on Harman and Finnigan, 2007, 2008) in the surface-

energy balance (SEB) of a one-dimensional single column atmospheric model, in order to study the effect of the RSL on the coupling between a canopy and the boundary layer. Based on their (Harman and Finnigan, 2007, 2008) RSL formulation, the roughness parameters (e.g. the roughness length of momentum and scalars, displacement plane) are stability dependent variables. Harman (2012) found an altered surface fluxes about 25 % (e.g. sensible heat flux and the friction velocity), and also effects on mean boundary state variables (e.g. wind speed, potential) just above the canopy when RSL is applied.

Extending these previous works, our study aimed to elucidate the ABL system for real conditions, taking the representation of the RSL into account. In order to consider all the relevant physical processes needed to represent the diurnal variability of the state variables above the canopy, we implemented the RSL formulation proposed by Harman and Finnigan (2007, 2008)



and embedded it in a coupled soil-vegetation-atmosphere mixed-layer model (van Heerwaarden et al., 2009). The model has been successfully employed in a number of studies based on field observations gathered above low vegetation (e.g. van Heerwaarden et al., 2009) or influenced by complex surface heterogeneity and topography (e.g. Pietersen et al., 2015). Here, we extend its applicability, by employing the RSL model (Harman and Finnigan, 2007, 2008) to study a surface with

relatively tall and sparse uniform plant canopy. In order to constrain and evaluate our numerical experiments, we took advantage of the comprehensive data set-gathered during the Canopy Horizontal Array Turbulence Study (CHATS) experiment (Patton et al., 2011), paying special attention to sensitivity analysis of the CBL dynamics to the scaling variables that govern the RSL parameterization. We focused on the sensitivity of the model results to changes in the canopy adjustment length scale, $L_c$, and the $\beta = u_*/|U|$ ratio at the canopy top, which are dependent on respectively the canopy

structure and atmospheric stability.

Our research is thus an exploratory study of the potential alterations to the boundary-layer dynamics as calculated by large-scale models (e.g. Chen and Dudhia, 2001), when the RSL is taken into account.

## 2 Methods

### 2.1 CHATS data

The CHATS experiment took place in the spring of 2007 in one of Cilker Orchards' walnut blocks in Dixon, California, USA. A detailed description of the site, instrumentation and data treatment has been provided by Patton et al. (2011) and Dupont and Patton (2012a). Here we focus on the specific observations used in this study and on the criteria used to select the representative cases.

The observations analysed in this study were made on a 30 m mast located near the northernmost border of the orchard site

in order to ensure a fetch of about 1.5 km for the predominant southerly winds (see Fig. 1a and Fig. 3 in Dupont and Patton, 2012a). The average height of the trees ($h_c$) was estimated to be 10 m. Wind, temperature and specific humidity were measured at 13 levels on the mast (see Patton et al., 2011). The shortwave and longwave radiation above the canopy were measured at 6 m above the canopy top. The soil properties were measured at a depth of 0.05 m . The NCAR Raman-shifted Eye-safe Aerosol Lidar (REAL) monitored reflectivity in order to evaluate the evolution of the boundary-layer height, $h$

(Patton et al., 2011). The Lidar measurements enabled us to retrieve the evolution of $h$ from the aerosol backscatter signal (see supplementary material for the method and the data treatment procedures). The leaf area index (LAI) was also measured before and after the growing (leaf-out) season (Patton et al., 2011). Although the LAI varied from 0.7 to 2.5 m$^2$ (leaf area) m$^{-2}$ (surface area) depending on the seasonality (before and after leaf-out, respectively), we took the value of 2.5 for the LAI to represent a fully vegetated canopy. It is important to note that due to the sparseness of the orchard canopy the insolation at

the ground was relatively high, leading to high available energy at the soil. In consequence, the soil-related fluxes of sensible





and latent heat were relatively important for the turbulent exchange processes within and above the canopy (Dupont and Patton, 2012b; Shapkalijevski et al., 2016).

The CHATS dataset is used in our study to initialise and constrain our soil-vegetation-atmosphere modelling system. The model evaluation of the diurnal variability of the state variables in and above the roughness sublayer makes use of diurnal observations of the mean and turbulent variables at the same heights (at the canopy top (10m) and at 19 m above the canopy) as for the selected study cases (Sect 2.3).

## 2.2 Soil-vegetation-atmosphere model

An atmospheric boundary-layer model with a zero-order jump approach, based on mixed-layer theory (Lilly, 1968; Tennekes and Driedonks, 1981; Vilà-Guerau de Arellano et al., 2015), was used to calculate the evolution of the well-mixed (slab) state variables and the evolution of boundary layer height. It is based on the vertical integration of the slab-averaged governing equations of thermodynamic variables and atmospheric constituents well above the canopy. At the upper boundary of the atmospheric model, the thermal inversion layer separates the well-mixed layer (MXL) from the free troposphere (FT). This separation is represented by a finite jump in the constituent under consideration (FT values minus MXL value) over an infinitesimal depth. At the bottom, we included a representation of the surface roughness sublayers (RSL), which is characterized by steep mean gradients, connecting the surface to the lower part of the surface layer (ASL). The ASL then connects the RSL to the MXL (Fig. 1).

*< place Figure 1 somewhere here >*

Based on the mixed-layer model, the diurnal variability of the mean thermodynamic variables and atmospheric constituents reads as follows:

$$\frac{d\langle\varphi\rangle}{dt} = \frac{(\overline{w'\varphi'})_s - (\overline{w'\varphi'})_e}{h} + Adv_\varphi, \tag{1}$$

where $\left(\overline{w'\varphi'}\right)_s$ and $\left(\overline{w'\varphi'}\right)_e$ are the vertical turbulent kinematic fluxes of a certain variable $\varphi$ ($\varphi \equiv u, v, \theta, q$) at the lower (surface) and upper (entrainment) boundaries, respectively; $h$ is the boundary-layer height, while $Adv_\varphi$ is the advection of the corresponding quantity of interest. The chevrons "$\langle\varphi\rangle$" represents the variables within the mixed layer. For a more complete description of the mixed-layer governing equations, see van Heerwaarden et al. (2009) and Ouwersloot et al. (2012). In what follows, we incorporate the most physically sound roughness-sublayer model (Harman and Finnigan, 2007, 2008) in the surface scheme of our modelling system (following the concept of Harman, 2012). We calculated the surface fluxes in Eq. (1) as follows:





$$\left(\overline{w'\varphi'}\right)_s = \frac{(\varphi_s - \varphi(z_r))}{r_{a\varphi}(z_r) + r_{s\varphi}}, \tag{2}$$

where $\varphi_s$ and $\varphi(z_r)$ are the mean vector (wind velocity) and scalar (potential temperature, specific humidity) quantities at roughness length $(z_{0\varphi})$ and at a given reference height within the RSL $(z_r)$. For momentum $z_{0\varphi} \equiv z_{0M}$, while for scalars $z_{0\varphi} \equiv z_{0H}$. The aerodynamic resistance in Eq. (2) is calculated at $z_r$ and is related to the drag coefficient $(C_\varphi)$ and the mean wind speed $(|U|)$ at the same height:

$$r_{a\varphi} = \left(C_\varphi(z_r)|U(z_r)|\right)^{-1}, \tag{3}$$

The stomatal resistance, $r_{s\varphi}$, in Eq. (2) is equal to zero for momentum and heat. Its definition and computation for moisture is presented and explained in van Heerwaarden et al. (2009).

The influenced $C_\varphi(z_r)$ and $\varphi(z_r)$ due to the canopy presence are calculated using the following expressions:

$$C_\varphi(z_r) = \frac{\kappa^2}{\left[\ln\left(\frac{z_r}{z_{0M}}\right) - \Psi_M\left(\frac{z_r}{L}\right) + \Psi_M\left(\frac{z_{0M}}{L}\right) + \widehat{\Psi}_M(z_r, d_t, L)\right]\left[\ln\left(\frac{z_r}{z_{0\varphi}}\right) - \Psi_\varphi\left(\frac{z_r}{L}\right) + \Psi_\varphi\left(\frac{z_{0\varphi}}{L}\right) + \widehat{\Psi}_\varphi(z_r, d_t, L)\right]}, \tag{4}$$

and

$$\varphi(z_r) = \varphi_s - \frac{\overline{(w'\varphi')}_s}{\kappa u_*}\left[\ln\left(\frac{z_r}{z_{0\varphi}}\right) - \Psi_\varphi\left(\frac{z_r}{L}\right) + \Psi_\varphi\left(\frac{z_{0\varphi}}{L}\right) + \widehat{\Psi}_\varphi(z_r, d_t, L)\right], \tag{5}$$

where $\kappa$ is the von-Kármán constant of 0.41 (Högström, 1996). The friction velocity is computed as:

$$u_* = \sqrt{C_M(z_r)|U(z_r)|}, \tag{6}$$

The functions: $\Psi_M\left(\frac{z_r}{L}\right)$, $\Psi_M\left(\frac{z_{0M}}{L}\right)$, $\Psi_\varphi\left(\frac{z_r}{L}\right)$, $\Psi_\varphi\left(\frac{z_{0\varphi}}{L}\right)$ are the integrated diabatic stability functions for momentum and scalars, while $\widehat{\Psi}_M(z_r, d_t, L)$ and $\widehat{\Psi}_\varphi(z_r, d_t, L)$ represent the roughness sublayer functions for momentum and scalars (Harman and Finnigan, 2007, 2008). Stability-dependent roughness lengths for momentum and other scalars ($z_{0M}$ and $z_{0\varphi}$, respectively) included in Eqs. (4 and 5) are described in detail in Harman (2012).

The displacement height, $d_t$, in Eqs. (4) and (5) is defined as the distance from the conventional displacement plane, at actual height $d$, to the canopy top, at actual height $h_c$: $d_t = h_c - d$ (see Fig. 1). Based on Harman and Finnigan (2007), $d_t$ is calculated as follows:

$$d_t = \beta^2 L_c, \tag{7}$$





where, $L_c$, is canopy adjustment length scale, defined as:

$$L_c = (c_d\, a)^{-1}, \tag{8}$$

where $a$ is the canopy's leaf area density which is assumed to be constant with height, while $c_d$ is the leaf drag coefficient
(Harman and Finnigan, 2007). The canopy adjustment length scale (Eq. 8) is defined as a measure of the distance over which

an internal boundary layer with no prior knowledge of a tall canopy would need to equilibrate (adjust) to the presence of a
canopy (Belcher et al., 2003; Harman and Finnigan, 2007). For the given CHATS experiment, Shapkalijevski et al. (2016)
have shown that $L_c = 16m$ under near-neutral and weakly unstable conditions. Under strong unstable conditions $L_c \approx 10m$,
while under strong stable conditions $L_c > 20m$. Another critical stability-dependent variable in Eq. (7) is $\beta$, which indicates
the ratio between the friction velocity and the mean wind speed at canopy top ($\beta = u_*/|U|$). Based on our CHATS analysis

(Shapkalijevski et al. 2016), we find that under weakly unstable, near-neutral and weakly stable atmospheric conditions $\beta$
has constant value of 0.3, consistent with Harman and Finnigan (2007, 2008). Under strong unstable conditions, this variable
increases up to 0.4, while under strong stable conditions it decreases to nearly 0.25. Based on estimates at the CHATS site,
we assume the values of 0.3 and 16m for $\beta$ and $L_c$, respectively. The sensitivity of the calculated surface fluxes and
boundary state variables to the values of $\beta$ and $L_c$ is presented and discussed in Sects. 4 and 5.

Finally, the RSL functions $\widehat{\Psi}_M(z_r, d_t, L)$ and $\widehat{\Psi}_\varphi(z_r, d_t, L)$, are non-linear integrals, which are solved numerically. For a
detailed theoretical description and derivation of these RSL functions, see Harman and Finnigan (2007, 2008).

**2.3 Research strategy**

To initialise and validate our modelling system, we selected observations of a representative day from the second phase of
the CHATS campaign (from 13 May to 12 June) focusing on the walnut trees after leaf-out (fully vegetated canopy). The

representative case is based on two requirements that the data satisfied: *i*) well-mixed conditions and *ii*) well-developed RSL.
Our assumption of a well-mixed boundary layer is justified for sunny (cloudless) days characterised by convective
conditions. Moreover, the LIDAR data (see figures in supplementary material) showed a quite homogeneous signal, which in
the absence of radiosoundings implies well-mixed conditions up to 500 m height at noon (12:00 LT). In order to ensure the
maximum influence (fetch) of the canopy on the atmospheric flow, leading to a potentially well-developed RSL, we selected

data with southerly predominant winds, since the measurement tower was placed at the northernmost part of the orchard
field (Fig. 1 of Patton et al., 2011). Based on these requirements, we selected observations from 27 May 2007 at CHATS. To
test the robustness of the model results, we also analysed an additional day (31 May 2007) with different wind forcing
(northerly varying to southerly winds in the course of the day).



Several systematic experiments were performed, in which the representation of the drag coefficient and the impact of the RSL on mean gradients (Eqs. 4-5), as well as the inclusion of various large-scale forcing were varied. The standard MOST runs (abbreviated as 'M') were performed by omitting the roughness sublayer functions in Eqs. (4)-(5). The toggled large-scale forcing consists of mean vertical velocity subsidence, advection of cold and moist air, and increased boundary layer drying due to a drier free troposphere (see next paragraph). Table 1 summarises the processes included in the numerical experiments.

*< place Table 1 somewhere here >*

The numerical experiment which does not take subsidence into account has prescribed zero subsidence (no divergence of the mean horizontal wind), while the numerical experiments with subsidence have imposed constant divergence of the mean horizontal wind (Appendix A1). Moreover, and based on the observed temporal evolution of the potential temperature and specific humidity at 29 m, we set constant advective cooling and moistening at specific moment in time in our numerical experiments (Appendix A1). No advection of momentum has been imposed in the momentum budget. Furthermore, to represent the increased BL drying from the free troposphere we modified the specific humidity lapse rate in the free troposphere ($\gamma_q$) depending on the BL-height (Appendix A1). For instance, to represent the observed temporal evolution of the specific humidity at 29 m during the day on 27 May 2007, we prescribed a modification of the $\gamma_q = 10^{-4}$ kg kg$^{-1}$ m$^{-1}$ when the BL-height reaches 450 m (based on observations), while the initial $\gamma_q$ was set equal to 0 units (see Table A1.1).

The numerical experiments started at 08:00 local time (LT), which is equivalent to 15:00 coordinated universal time (UTC), and lasted for nine hours. In the absence of initial measurements at the residual layer (roughly 350 m); we imposed the upper boundary conditions of the model to optimise the representation of the temporal evolution of the potential temperature, specific humidity, wind direction and boundary-layer height (Table A1.1 and A1.2 in Appendix A1). We used the observations at the highest measurement level at the tower (29 m above ground surface) to evaluate the model results away from the canopy, where the RSL effects are minimal.

Furthermore, we put special emphasis on validating the modelled quantities at the canopy top ($z = z_r = d_t$) and compared them with the corresponding observations at the same height. We selected the canopy top (10 m above the ground surface) as a reference level due to the largest expected RSL effects on the flow (Harman and Finnigan, 2007, 2008). We note that the area of the orchard is rather small (~1 km$^2$) to be capable of influencing the development of the boundary-layer dynamics (Schmid, 2002). However, in the model, we extrapolated the characteristic surface fluxes and mean gradients, assuming that the area of this orchard is sufficient to drive the main processes at the CBL dynamics.

Finally, the initial value of $z_{0M} = 0.7$m used in all the numerical runs (Appendix A1.1) was estimated based on the approach developed by Raupach (1994) for a LAI of 2.5 and $\beta = 0.3$. Thus, the initial value of the roughness length for scalars,





$z_{0M} = 0.095\text{m}$ (see Table A1.1 in Appendix A1), is calculated as $\ln\left(\frac{z_{0M}}{z_{0H}}\right) = 2$ (see Physick and Garratt, 1995). For the standard MOST runs (MXL+MSAD), we used invariant (fixed) $z_{0M}$ and $z_{0\varphi}$ with values equal to their corresponding initial values, while when including the RSL, we used stability dependent formulation for $z_{0M}$ and $z_{0\varphi}$ (Harman and Finnigan, 2007, 2008).

## 3 Model validation

### 3.1 Radiation and surface energy balance

We start our analysis by evaluating the modelling system to represent the observations of the selected study cases. Figure 2a,b shows the observed and modelled components of the net radiation: downwelling (↓) and upwelling (↑) shortwave (SW) and longwave (LW) radiation fluxes above the canopy (measured at 6 m above the canopy top). The various radiation components are well reproduced by the model.

Figure 2c,d shows the four terms of the surface energy balance (Rn = SH + LE + G) for both cases, respectively. While the net radiation fluxes compare satisfactorily with the observations, the modelled daily averaged values of SH and LE are overestimated: 30% and 15% larger than the observed LE and SH, respectively for both case studies (27 and 31 May 2007). The average daily difference in the modelled and observed ground flux is up to 5 W m$^{-2}$. The diurnal variations in the observed LE and SH are well captured by the model, for instance the rapid decay of SH towards the end of the day relative to LE.

*< place Figure 2 somewhere here >*

Our explanation of this overestimation is the frequently observed imbalance of the observed surface energy system (Foken, 2008). This hypothesis is corroborated by an observed daily average difference of up to -30% of SH + LE + $\Delta Q_s$ compared to Rn-G for the case of 27 May and -20% on 31 May (Fig. 3), even when the heat storage contribution ($\Delta Q_s$) is included in the observed SEB (up to 5% energy input in the total balance). The $\Delta Q_s$ is the sum of the sensible ($\Delta Q_a$) and latent ($\Delta Q_w$) heat storage in the air column (including the canopy space) below the flux measurements by eddy-covariance (EC). The method used to calculate $\Delta Q_s$ from the observed potential temperature and specific humidity at the levels within and above the canopy, but below the height of EC observations, is based on that described by McCaughey and Saxton (1988) and later used in Oliphant et al. (2004). The heat stored in the biomass and the energy used in the photosynthesis are neglected in our case, since according to Thom et al. (1975), Ohta et al. (1999) and Jacobs et al. (2007) these two terms are negligibly small (less than 2 % of total Rn). The values of the surface energy imbalance at CHATS are similar to those found by a number of other observational studies, showing an average of up to 20% surface energy imbalance, as listed in Section 3.7 of Foken





(2008). With regard to our own research, it is important to note that related to this non-closure of the observed SEB, the observed SH and LE are too low, so the modelled SH and LE are more likely to be the correct values.

*< place Figure 3 somewhere here >*

The comparison presented here confirms that our modelling system is capable of reproducing the diurnal variations in radiation and surface energy balance with sufficient accuracy to reproduce the diurnal variations in the local state variables, as the following sections describe.

## 3.2 CBL dynamics

Figure 4 shows the observed and modelled diurnal evolution of the boundary-layer height, mixed-layer potential temperature and specific humidity for the case of 27 May 2007. The boundary-layer height (Fig. 4a), $h$, increases during the morning hours from 350m to up to 500m at around 11:00 LT, after which $h$ remains almost constant before it starts to decay at around 14:00 LT. In the absence of data on the vertical profiles of potential temperature and specific humidity in the mixed layer and the entrainment zone, we are unable to judge whether this more rapid growth until 11:00 LT is due to a progressive growth of the CBL into a residual layer above the canopy (Ouwersloot et al., 2012). Since our aim is to study the RSL effects on CBL dynamics, here we focus our analysis to the numerical experiments described above.

It is important to mention that $h$, as observed by the LIDAR backscatter data, is very sensitive to the morning-noon transition (08:00 - 10:00 LT) and late afternoon-evening (after 16:00 LT) transition conditions. This is due to possible non-uniform backscatter profiles, which can contain multiple maximum gradients, impairing the ability of the automated method to retrieve $h$ (see supplementary material). Therefore, the accuracy of the observations of $h$ is better under well-mixed conditions (from 10:00 to 16:00 LT in our case). During this period, only the model runs that take into account the subsidence and advective cooling (MXL+RSAD and MXL+RSA) capture the evolution (relatively steady) of the $h$ sufficiently well after the morning transition (Fig. 4a, in connection with Table 1). This result implies a significant influence of the subsidence, and to a lesser extent the effects of advective cooling, on boundary-layer growth for the given case. Figure 4a also shows that the effect of the RSL on the evolution of $h$ is insignificant (MXL+RSAD vs MXL+MSAD).

*< place Figure 4 somewhere here >*

The role of the large-scale advective cooling on the CBL dynamics was also recorded through the diurnal evolution of the potential temperature (Fig. 4b). Between 10:00 LT and 12:00 LT, a non-local advective cooling process resulted in a slowdown in the increase of the potential temperature. We hypothesise that the rapid cooling before noon is related to the advection of cold air, probably due to a sea-breeze front, which is frequently observed at the CHATS site (Mayor, 2011). We took this process into account in our numerical experiment (MXL+RSA) by imposing a constant advection of cold air





between 10:00 LT and 17:00 LT (Table 1). The strength of the advective cooling in the model was arbitrarily chosen to provide the best representation of the observed mixed-layer quantities (Table A1.1, Appendix A1). As Fig. 4b shows, while taking only surface forcings, entrainment processes and subsidence into account does not suffice to represent this case (experiment MXL+RS), the potential temperature evolution is captured well if the advection is taken into account
(experiment MXL+RSA) as well.

Similar behaviour of the diurnal evolution of the specific humidity at 29m above the ground surface was observed (Fig. 4c). Here, the large-scale advective process is displayed by a significant jump in the magnitude of the specific humidity (from 7.9 g kg$^{-1}$ to as much as 8.5 g kg$^{-1}$) immediately after 10:00 LT. In the absence of observed specific-humidity profiles, we hypothesise that this increase in moisture content is due to an air mass transported by the sea-breeze front coming from the
bay area (east and southeast). It is also possible that during the morning transition this sudden change is caused by the existence of a residual layer, which becomes connected to a growing shallow layer (Ouwersloot et al., 2012). However as mentioned before, since there are no data to explain the latter, but also because main focus of this study is the effects of the RSL on the CBL dynamics, we limited our analysis to the numerical experiments described above. After this increase, $q$ remains almost constant on time until the end of the day (17:00 LT). This is probably related to the drying associated with
the entrainment of free tropospheric (drier) air into the boundary layer. Based on the observed $q$ in the hours after 11:00 LT, the transport of dry air from the free troposphere is dominant, preventing the rise in the specific humidity, which results in a relatively constant value. The diurnal evolution of the specific humidity is well represented by the model run that takes the subsidence, advection and drying from the free troposphere into accounts (MXL+RSAD).  On the other hand, the model runs which do not take the drying (MXL+RSA) and the advection and drying (MXL+RS) into account overestimate the specific
humidity after 11:00 LT.

The analysis presented in Fig. 4 shows that the complex boundary-layer structure at the CHATS site is highly dependent on the large-scale effects, including subsidence, advective cooling and moistening, as well as entrainment of dry air from the free troposphere.

The observed diurnal variability of the wind enables us to further verify the role of the large-scale forcing and the local
canopy. Here, we compare the observed and modelled temporal evolution of the wind direction, individual wind speed components and absolute wind velocity (Fig. 5). The model is well able to represent the observed temporal evolution of wind, except for the period between 10:00 and 11:00 PLT, when outliers are present in the observed wind components (Fig. 5c) and, consequently, the wind direction (Fig. 5a). These outliers are associated with the sharp changes in the wind forcing (northerly winds present between 10:00 and 11:00 LT), a phenomenon observed daily before noon throughout whole
campaign (based on observed time series) (see also Zaremba and Carroll, 1999). Combining the individual wind components closely approximates the wind speed, which displays an almost constant acceleration during the day (Fig. 5b) and (after 11:00 LT) an almost constant friction velocity (see Figure 6c).




*< place Figure 5 somewhere here >*

The results of the case study of 27 May 2007 are corroborated by those of the case study of 31 May 2007 (not shown), showing similar patterns and structure of the CBL dynamics in both cases.

In summary, our modelling system is capable of reproducing the land-canopy-atmosphere characteristics of the case studies with satisfactory accuracy at a height well above the canopy. In the following section, we study the impact of the canopy on the boundary-layer state variables within the roughness sublayer near the canopy top.

## 4 The wind in the RSL and its effect on the bulk momentum budget

Figure 6 shows the observed and modelled temporal evolution of the mean wind speed, drag coefficient and friction velocity at the canopy top. The numerical experiment MXL+RSAD of the coupled modelling system satisfactorily represents the evolution of the wind at this level, while omitting the RSL effects (MXL+MSAD) results in underestimation of the wind speed (reaching a daily average of up to 50 %; Fig. 6a). This is in agreement with previous studies based on comparisons of observed and modelled wind profile (Physick and Garratt, 1995; Harman and Finnigan, 2007). The main effect of the canopy is a modification of the drag. Omitting the RSL effects (MXL+MSAD vs MXL+RSAD) results in significant overestimation of $C_M$ by a factor of up to four (Fig. 6b), in accordance with the analysis provided by De Ridder (2010).

*< place Figure 6 somewhere here >*

Both the MXL+RSAD and MXL+MSAD model runs, i.e., with and without the effects of the RSL included, underestimate $u_*$ by about 20 % (Fig. 6c). Like Physick and Garratt (1995), we found small RSL effects on the modelled friction velocity in the case studies (6 %). The similarity between the friction velocities is due to compensating effects of the drag coefficient and the wind speed modulus (Eqs. 4-6). Both $C_M$ and $|U|$ are altered in opposite directions, with magnitudes that fit the observation (Fig. 6a,b), thus leading to a relatively constant $u_*$ (Fig. 6c).

In order to extend and generalise our results, we performed a parameter-space sensitivity analysis on two stability-dependent scales in the RSL formulation: $L_c$ and $\beta$ (see also Sect. 2.2). Figure 6d summarises the results of the sensitivity analysis at 13:00 LT. The variations in $\beta$ ($0.25 \leq \beta \leq 0.4$) and $L_c$ ($10 \leq L_c \leq 20$ m) have a significant impact on $z_{0_M}$ and $u_*$, but a relatively small impact on $h$. We find that $u_*$ is sensitive to the changes in $\beta$ and $L_c$ with a maximum variation at 13:00 LT of up to 25 % ($0.29 \leq u_* \leq 0.37$ m s$^{-1}$) with respect to the case study value ($u_* = 0.32$ m s$^{-1}$) for the range of conditions investigated here. In our analysis, varying these scales, dependent on stability (based on the CHATS data), results in $h$ variation of up to 6 % (Fig. 6d).




We further extend our analysis of the impact of the canopy-related parameters on the atmospheric flow by studying their relative contribution to the momentum budget, compared to other contributions, e.g. entrainment or geostrophic forcing (Appendix A2). For this, we keep $L_c$ equal to 16 m and in the first experiment, we set $\beta = 0.25$ (typical for more stratified conditions), while in the second experiment we set $\beta = 0.40$ (typical for unstable conditions). Varying $L_c$ did not yield relevant differences in the wind budget (not shown).

Figure 7 shows that on average the momentum tendency due to surface stress is approximately 25 % larger for $\beta = 0.40$ than when $\beta = 0.25$. This enhanced tendency is partially compensated for by an increase in geostrophic forcing through the whole day and, to a lesser degree, entrainment. This results in a similar total momentum tendency in both cases.

Figure 7 also shows the tendencies of the three components of the total wind-speed budget (Appendix A2): surface forcing, the momentum entrainment, and the geostrophic forcing. The surface forcing, combining the surface stress and canopy drag, always leads to a negative tendency in the momentum, while entrainment from free-tropospheric air results in a positive tendency. In the case under study, the tendencies of the ageostrophic components are also usually positive. The resulting total momentum tendency is positive after 09:00 LT (Fig. 7).

*< place Figure 7 somewhere here >*

**5 Heat and moisture**

The impact of the RSL on the potential temperature and specific humidity at canopy-top level and their respective surface heat fluxes is presented in Fig. (8) and Fig. (9). Here, similar analyses were performed as for momentum in the previous section. The modelled potential temperature at this level is in good agreement with the observations. The suppressed increase in potential temperature before noon is caused by the large-scale advective cooling that sets in after 10:00 LT. The MXL+RSAD model run, including the RSL effects, performs better than the MXL+MSAD with differences of up to 1 K. Furthermore, the sensitivity analysis performed by varying $L_c$ and $\beta$ (Sect. 2.2) shows that $\theta$ differs by up to almost 1 K at 13:00 LT for the selected sensitivity ranges (Fig. 8b). For the same time, the sensible heat flux ranges between 302 and 306 W m$^{-2}$ (or less than 2 % with respect to the case study value at 13:00 LT).

*< place Figure 8 somewhere here >*

We find a slightly larger disagreement in the results for observed and modelled specific humidity at canopy-top level (up to 0.5 g kg$^{-1}$, or around 5 % with respect to the observed values). An interesting feature of the observations is the small difference in the magnitude (no greater than 0.5 g kg$^{-1}$) between 29 m above ground (Fig. 4c) and canopy top (Fig. 9a), but we were not able to explicitly explain this small difference in $q$ between these two levels. Like the potential temperature, $q$ is sensitive to $L_c$ and $\beta$ at 13:00 LT, with $q$ ranging from 10.3 g kg$^{-1}$ under unstable conditions to 9.0 g kg$^{-1}$ under weakly





stable conditions. The maximum variations in LE for different $L_c$ and $\beta$ is around 34 W m$^{-2}$, or around 9 % with respect to the case study value at 13:00 LT (362 W m$^{-2}$).

*< place Figure 9 somewhere here >*

Finally, in the range of $L_c$ and $\beta$ investigated, we found that the effective displacement height ($d_t$) can range from less than 5   1m to up to 3m meters (Eq. 7, Fig. 9b). This significantly affects the roughness lengths for momentum and scalars, since $z_{0M}$ and $z_{0\varphi}$ are directrly dependent on $d_t$ and stability (Harman and Finnigan, 2007, 2008; Zilitinkevich et al., 2008). These variations in the displacement height and the roughness lengths (Fig. 6d and Fig. 8b) are the cause of the variations in the surface fluxes (e.g. 2 % variation in SH and 9 % variation in LE).

**6 Discussion**

10   The interpretation of the CHATS height-dependent observations, employing a numerical model that integrates various spatial-temporal scales relevant within the CBL, reveals that the diurnal variability of the state variables above the orchard canopy is highly dependent on the contributions of local and non-local effects. Local effects are related to the land-canopy-atmosphere exchange of momentum and energy, while the non-local effects are either driven by boundary-layer dynamics, such as entrainment, or by mesoscale phenomena, such as subsidence and/or horizontal advection.

15   At meso-scales, as described by Hayes et al. (1989), Zaremba and Carroll (1999), Bianco et al. (2011) and Mayor (2011) the CHATS site is strongly influenced by various interacting mesoscale flows such as marine fronts and mountain-valley flows. Since this study focuses on convective conditions, and following the classification suggested by Zaremba and Carroll (1999, Table 3 and Fig. 4b,c), we studied two cases characterized by different mesoscale circulations: *i*) a case with southerly dominant winds and *ii*) a day with northerly winds that veer south at around noon. In both cases, the impact of the marine 20   mesoscale flow coming from the San Francisco Bay area (e.g. Fig. 7b,c in Zaremba and Carroll, 1999) leads to a sudden decrease in the rate of growth of the boundary-layer height (Fig. 4a). This yielded an almost constant $h$ at around 500m for the case of 27 May (Fig. 4a) and around 650 m on 31 May (see supplementary material).

In the absence of detailed observations of the temporal evolution at the entrainment zone, we are able to provide only first-order estimates of the large-scale effects relevant to our cases and discuss their impacts on the budgets of potential 25   temperature and specific humidity (Fig. 10). The budgets of potential temperature (Fig. 10a) and specific humidly (Fig. 10b) enable us to quantify the relevance of non-local versus local processes. Overall, surface and entrainment are the main contributors to the variability of the potential temperature and specific humidity. Besides these, the advective cooling and moistening process has a relatively large impact on the corresponding budgets after 10:00 LT, when advection is employed to capture the observed diurnal evolution of $\theta$ and $q$ (Fig. 4b,c). The  negative $\theta$-tendency and positive $q$-tendency due to





advection in this analysis (the green solid lines in Fig. 10) corroborate the drop in air temperature and increase in moisture which were observed over the Sacramento Valley flow, characterised by southerly winds (Zaremba and Carroll, 1999; Bianco et al., 2011).

*< place Figure 10 somewhere here >*

Focussing now on the surface conditions, and on canopy scales, the representation of the RSL has a large impact on the drag coefficients and mean gradients of the thermodynamic variables within the RSL, and to a lesser extent to the surface fluxes. Our findings are in agreement with those of Physick and Garratt (1995) and Maurer et al. (2013), and raise a potential paradox. Even though surface fluxes inferred from gradient observations just above the canopy are affected by roughness sublayer effects (e.g., Mölder et al., 1999; De Ridder, 2010), the actual (modelled) fluxes are only insignificantly different

for the standard conditions ($L_c = 16$ m and $\beta = 0.3$). This is due to the parameterisation of the surface fluxes depending on both the drag coefficient and the difference of the mean variable (Eqs. 2-3). As we showed (e.g. Fig. 6a,b), both are strongly affected by the effects of RSL correction, but they compensate each other. The momentum flux is more sensitive to the variations in $L_c$ and $\beta$ than the sensible and latent heat fluxes. This is due to the boundary condition that relates the surface value to the atmospheric value. While a Dirichlet boundary condition is applied to momentum (no wind at roughness height

for standard MOST), a Neumann boundary condition is required for potential temperature and specific humidity. $T_s$ depends on the SEB (Sect. 2.2,) and is determined as a function of the radiation, soil heat flux, $\theta(z_r)$, $q(z_r)$, $r_{aH}$ and $r_s$ (see e.g. van Heerwaarden et al., 2009). Since $\theta(z_r)$, $q(z_r)$, $r_{aH}$ and $r_s$ are altered by the RSL, $T_s$ and $q_s$ are affected as well, resulting in minor variations in the mean gradient (see also Harman, 2012, Fig. 4a,b) and therefore smaller variation in the surface flux (Eqs. 2). This is the reason why we found larger fluctuation in the friction velocity (25 %) for different RSL

scales ($\beta$ and $L_c$), compared to the much smaller variations in SH (2 %) and LE (9 %).

**7 Conclusions**

By combining observations, collected at different heights above a walnut orchard canopy during the Canopy Horizontal Array Study (CHATS), with model experiments performed incorporating a land-vegetation-atmosphere model, we investigated the contributions of canopy and large-scale atmospheric forcings on the diurnal variability of boundary-layer

height, the evolution of mixed-layer properties and of canopy-atmosphere exchange of momentum, potential temperature and specific humidity. We selected a representative day with southerly wind conditions for our study to maximize the effects of the canopy fetch and compared it with another day ( wind veering from northerly to southerly) characterized by less fetch influence. We pay particular attention to determine the sensitivity of the surface fluxes and the boundary-layer evolution to changes in the canopy adjustment length scale, $L_c$, and the ratio between the friction velocity and the wind speed at the

canopy top, $\beta$, which are relevant scales within the roughness sublayer.





On the bases of our findings, we reach the following conclusions:

- The investigated CHATS convective boundary layers are strongly affected by large-scale processes such as advective cooling, subsidence and entrainment of dry and warm air from the free troposphere. Quantifying these large scaling forcings by using the observations, the coupled soil-vegetation-atmosphere modelling system satisfactorily represents the surface fluxes and convective boundary-layer dynamics at the CHATS site.

- By applying the roughness sublayer formulations within the surface scheme of the model, the representation of the diurnal evolution of the boundary layer state variables and the corresponding drag coefficients at the canopy height is improved. The drag coefficients and the mean gradients of the state variables at the canopy height change strongly when the new formulation, including the roughness sublayer, are applied. However, due to compensation between the drag coefficients and the differences in the mean variables at two levels within the roughness sublayer, the modelled surface momentum and heat fluxes remain relatively unchanged ($< 3\,\%$).

- The sensitivity analysis on roughness sublayer scales, analysed through changes in $L_c$ and $\beta$, and their diabatic stability dependence, led to changes in the friction velocity (up to 25 %) and smaller variations in the sensible and latent heat fluxes (2 % and 9 % respectively), leading to changes in the boundary layer height of up to 6 %.

- Changes in $\beta$ significantly impact the surface drag contribution to the mixed-layer momentum budget (up to 25 % variation for the given range of $\beta$). The altered surface momentum due to changes in $\beta$ is compensated by changes in geostrophic forcing and entrainment resulting in a similar total momentum tendency.

- When interpreting the CHATS measurements above the canopy, the mesoscale advective processes or subsidence play an important role in determining the convective boundary-layer dynamics. Analysis of the bulk potential temperature and specific humidity budgets showed that the influence of the advection can be around one fourth of the total potential temperature budgets.

Overall, the canopy's impact on convective boundary-layer dynamics is relatively insignificant, due to its small effect on surface fluxes and the bulk boundary-layer properties well above the canopy ($z > 2h_c$). However, the roughness sublayer parameterisation should be applied when comparing observations (e.g. tower measurements) and large-scale model outputs of the mean quantities and turbulent transfer coefficients near and just above the canopy, since it improves predictions of those quantities. These could potentially affect modelled emissions and deposition of chemical species by plant canopies, since these are dependent on local atmospheric conditions (Foken et al., 2012).





**Acknowledgments:** We would like to thank Dr. Ian Harman (CSIRO - Commonwealth Scientific and Industrial Research Organisation, Canberra, Australia) for providing us with the roughness sublayer model code, as well as Dr. Edward G. Patton (NCAR - National Center for Atmospheric Research, Boulder, Colorado) for giving us the access to the CHATS data set and the comments on the boundary-layer height evaluation.

5   **Appendix A1: Mixed-layer model initial and boundary conditions for two study cases at CHATS**

**Table A1.1.** Initial and boundary conditions for model runs of 27 May 2007 (147 DOY) for the CHATS experiment.

| Variable | Description and unit | value |
|---|---|---|
| | **MXL model run** | |
| $t$ | time domain [s] | 32 400 |
| $dt$ | time step [s] | 10 |
| lat | latitude [deg] | 38.45 N |
| lon | longitude [deg] | -121.8 E |
| DOY | day of the year | 147 |
| hour | starting time of the model run [LT] | 08:00 |
| | | |
| | **Boundary layer dynamics** | |
| $P_0$ | surface pressure [Pa] | 102900 |
| $h_0$ | boundary-layer height at 08:00 LT [m] | 350 |
| $Div_{|U|}$ | divergence of the mean horizontal wind [s$^{-1}$] | $5 \times 10^{-5}$ |
| $\langle\theta\rangle_0$ | initial mixed-layer potential temperature [K] | 286.5 |
| $\Delta\theta_0$ | initial temperature jump at entrainment zone [K] | 1.5 |
| $\gamma_\theta$ | potential temperature lapse rate in free troposphere [K m$^{-1}$] | 0.017 |
| $Adv_\theta$ | advection of heat [K s$^{-1}$](hour > 10:00 LT) | $0\ (-3 \times 10^{-4})^1$ |
| $\langle q\rangle_0$ | initial mixed-layer specific humidity [kg kg$^{-1}$] | $7.6 \times 10^{-3}$ |
| $\Delta q_0$ | initial specific humidity jump at entrainment zone [kg kg$^{-1}$] | $1.5 \times 10^{-4}$ |
| $\gamma_q$ | specific humidity lapse rate in free troposphere [kg kg$^{-1}$ m$^{-1}$] ($h > 450$ m) | $0\ (1 \times 10^{-4})$ |
| $Adv_q$ | advection of moisture [kg kg$^{-1}$ s$^{-1}$] (hour > 10:00 LT) | $0\ (2 \times 10^{-4})$ |
| $\langle u\rangle_0$ | initial longitudinal mixed-layer wind speed [m s$^{-1}$] | 0 |

---

[1] The values in the round brackets represent the prescribed changes in the model initialization depending on the boundary layer height (for $\gamma_\theta$ and $\gamma_q$) (if $h > 450$ m) and the time after 10:00 LT (for the advection).





| | | |
|---|---|---|
| $\langle v \rangle_0$ | initial lateral mixed-layer wind speed [m s$^{-1}$] | 1.5 |
| $\langle u \rangle_g$ | geostrophic longitudinal wind speed [m s$^{-1}$] | 0 |
| $\langle v \rangle_g$ | geostrophic lateral wind speed [m s$^{-1}$] | 4 |
| $\gamma_u$ | free atmosphere wind speed (longitudinal) lapse rate [s$^{-1}$] | 0.003 |
| $\gamma_v$ | free atmosphere wind speed (lateral) lapse rate [s$^{-1}$] | 0 |

**Roughness sublayer**

| | | |
|---|---|---|
| $z_{0M}$ | initial roughness length for momentum [m] | 0.7 |
| $z_{0\varphi}$ | initial roughness length for heat and moisture [m] | 0.095 |
| $L_c$ | roughness-sublayer penetration depth [m] | 16 |
| $\beta$ | roughness-sublayer scaling parameter [-] | 0.3 |

**Soil and vegetation**

| | | |
|---|---|---|
| $cc$ | cloud cover [-] | 0.07 |
| $\alpha$ | albedo [-] | 0.15 |
| $T_s$ | initial surface temperature [K] | 291 |
| $w_{wilt}$ | wilting point [m$^3$ m$^{-3}$] | 0.171 |
| $w_2$ | volumetric water content deeper soil layer [m$^3$ m$^{-3}$] | 0.26 |
| $w_g$ | volumetric water content top soil layer [m$^3$ m$^{-3}$] | 0.26 |
| $w_{fc}$ | volumetric water content field capacity [m$^3$ m$^{-3}$] | 0.323 |
| $w_{sat}$ | saturated volumetric water content [m$^3$ m$^{-3}$] | 0.472 |
| $C_{1sat}$ | coefficient force term moisture [-] | 0.132 |
| $C_{1res}$ | coefficient restore term moisture [-] | 1.8 |
| $rs_{min}$ | minimum resistance of transpiration [s m$^{-1}$] | 110 |
| $rs_{soil\_min}$ | minimum resistance of soil transpiration [s m$^{-1}$] | 50 |
| LAI | leave area index [m$^2$ m$^{-2}$] | 2.5 |
| $c_{veg}$ | vegetation fraction [-] | 0.9 |
| $T_{soil}$ | initial temperature top soil layer [K] | 290 |
| $T_2$ | temperature deeper soil layer [K] | 289 |
| $\Lambda$ | thermal conductivity skin layer divided by depth [W m$^{-2}$ K$^{-1}$] | 6 |
| $CG_{sat}$ | saturated soil conductivity for heat [K m$^2$ J$^{-1}$] | $3.6 \times 10^{-6}$ |



**Table A1.2.** Initial and boundary conditions for model runs of 31 May 2007 (151 DOY) for the CHATS experiment (similar to Table A1; here, only the differences are presented).

| Variable | Description and unit | value |
|---|---|---|
| | **Boundary layer dynamics** | |
| $h_0$ | boundary-layer height at 08:00 LT [m] | 250 |
| $Div_{|U|}$ | divergence of the mean horizontal wind [ $s^{-1}$] | $3 \times 10^{-5}$ |
| $\langle\theta\rangle_0$ | initial mixed-layer potential temperature [K] | 286.5 |
| $\Delta\theta_0$ | initial temperature jump at entrainment zone [K] | 1 |
| $\gamma_\theta$ | potential temperature lapse rate in free troposphere [K m$^{-1}$] | 0.017 |
| $Adv_\theta$ | advection of heat [K s-1] (hour > 10:00 LT) | $0\ (-1.3 \times 10^{-4})$ |
| $\langle q\rangle_0$ | initial mixed-layer specific humidity [kg kg$^{-1}$] | $7.6 \times 10^{-4}$ |
| $\Delta q_0$ | initial specific humidity jump at entrainment zone [kg kg$^{-1}$] | $1 \times 10^{-4}$ |
| $\gamma_q$ | specific humidity lapse rate in free troposphere [kg kg$^{-1}$ m$^{-1}$] ( $h > 500$ m) | $0\ (8 \times 10^{-5})^2$ |
| $\langle u\rangle_0$ | initial longitudinal mixed-layer wind speed [m s$^{-1}$] | 1 |
| $\langle v\rangle_0$ | initial lateral mixed-layer wind speed [m s$^{-1}$] | -2.0 |
| $\langle u\rangle_g$ | geostrophic longitudinal wind speed [m s$^{-1}$] | 0 |
| $\langle v\rangle_0$ | geostrophic lateral wind speed [m s$^{-1}$] ( $h > 500$ m) | -2(1.5) |
| $\gamma_u$ | free atmosphere wind speed (longitudinal) lapse rate [s$^{-1}$] | 0.008 |
| $\gamma_v$ | free atmosphere wind speed (lateral) lapse rate [s$^{-1}$] | 0 |

**Appendix A2: Momentum budget**

5    Assuming that in the free troposphere the wind is in balance (equilibrium) between the pressure gradients and Coriolis force, the budgets of the mixed-layer wind components are expressed by the following equations:

$$\frac{d\langle u\rangle}{dt} = \frac{\overline{w'u'}_s}{h} - \frac{\overline{w'u'}_e}{h} - f_c\left(\langle v\rangle - v_g\right), \tag{A2.1}$$

$$\frac{d\langle u\rangle}{dt} = \frac{\overline{w'u'}_s}{h} - \frac{\overline{w'u'}_e}{h} + f_c\left(\langle u\rangle - u_g\right), \tag{A2.2}$$

---

[2] The values in the round brackets represent the prescribed changes in the model initialization depending on the boundary layer height (for $\gamma_\theta$ and $\gamma_q$) (if $h > 500$ m) and the time after 11:00 LT (for the advection).





The modulus of the wind speed components is:

$$\langle |U| \rangle = \sqrt{\langle u \rangle^2 + \langle v \rangle^2} \, , \tag{A2.3}$$

Combing the Eqs. (A2.1 – A2.3), results in:

$$\frac{d\langle |U| \rangle}{dt} = \frac{1}{\langle |U| \rangle} \left\{ \left[ \left( \langle u \rangle \frac{\overline{w'u'}_s}{h} + \langle v \rangle \frac{\overline{w'v'}_s}{h} \right) - \left( \langle u \rangle \frac{\overline{w'u'}_e}{h} + \langle v \rangle \frac{\overline{w'v'}_e}{h} \right) \right] + f_c \left[ \langle v \rangle (\langle u \rangle - u_g) - \langle u \rangle (\langle v \rangle - v_g) \right] \right\} , \tag{A2.4}$$

where:

$\frac{d\langle |U| \rangle}{dt}$ is the *total wind speed* tendency;

$\frac{1}{\langle |U| \rangle} \left[ \left( \langle u \rangle \frac{\overline{w'u'}_s}{h} + \langle v \rangle \frac{\overline{w'v'}_s}{h} \right) \right]$ is the *surface forcing* (due to surface stress and canopy drag);

$\frac{1}{\langle |U| \rangle} \left[ - \left( \langle u \rangle \frac{\overline{w'u'}_e}{h} + \langle v \rangle \frac{\overline{w'v'}_e}{h} \right) \right]$ is the *entrainment forcing*;

$\frac{1}{\langle |U| \rangle} f_c \left[ \langle v \rangle (\langle u \rangle - u_g) - \langle u \rangle (\langle v \rangle - v_g) \right]$ is the *geostrophic forcing*.

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

**Tables:**

**Table 1. Numerical model runs; description and abbreviations.**

| Experiment abbreviation | RSL (R) or MOST (M) | Subsidence (S) | Advection (A) | FT drying (D) |
|---|---|---|---|---|
| **MXL+R** | R | - | - | - |
| **MXL+RS** | R | S | - | - |
| **MXL+RSA** | R | S | A | - |
| **MXL+RSAD** | R | S | A | D |
| **MXL+MSAD** | M | S | A | D |





**Figures:**

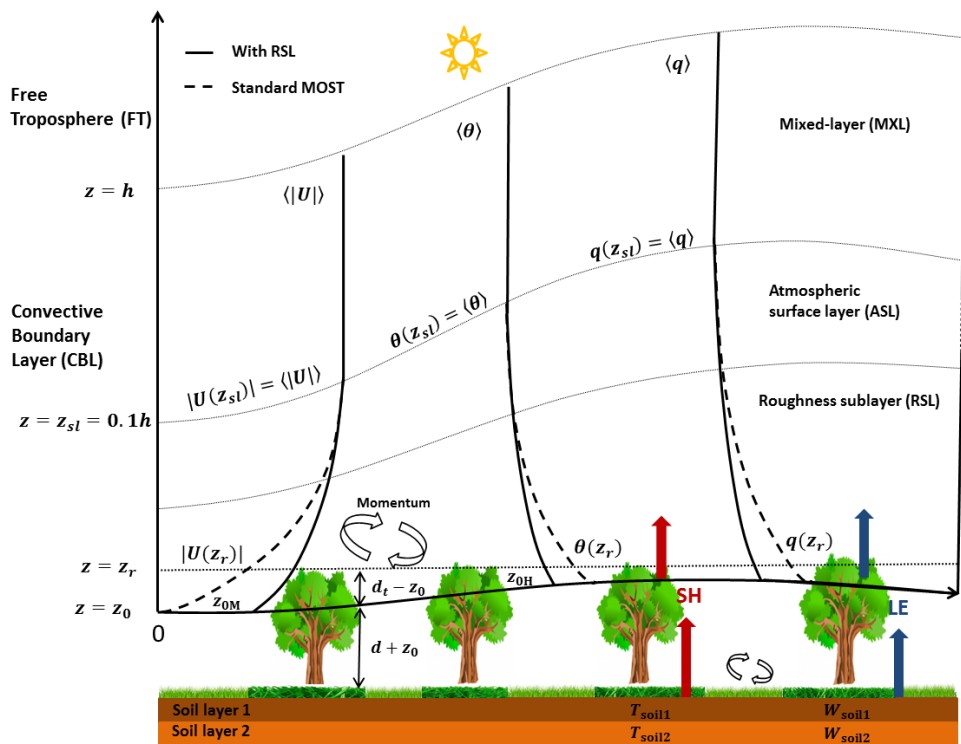

**Figure 1: Schematic overview of the coupled land-vegetation-atmospheric mix-layer model, with both including and omitting the RSL effects in the flux-profile relations. The vertical origin of the co-ordinate system is placed at the displacement height $d$. The height of the surface layer is calculate as 10 % of the boundary-layer height (Stull, 2009).**





**Figure 2: Observed and modelled radiation and surface energy balance components: (a) and (b) diurnal evolution of downwelling shortwave radiative flux (SW↓), upwelling shortwave radiative flux (SW↑), downwelling longwave radiative flux (LW↓) and upwelling longwave radiative flux (LW↑); (b) and (c) diurnal evolution of sensible heat flux (SH), latent heat flux (LE), the ground flux (G) and net radiation (Rn) (all in W m$^{-2}$). Observed quantities are**





**measured at 6 m above the canopy top. LT is local time (UTC-7). Sunrise was at 06:30 LT and sunset was at 19:30 LT.**

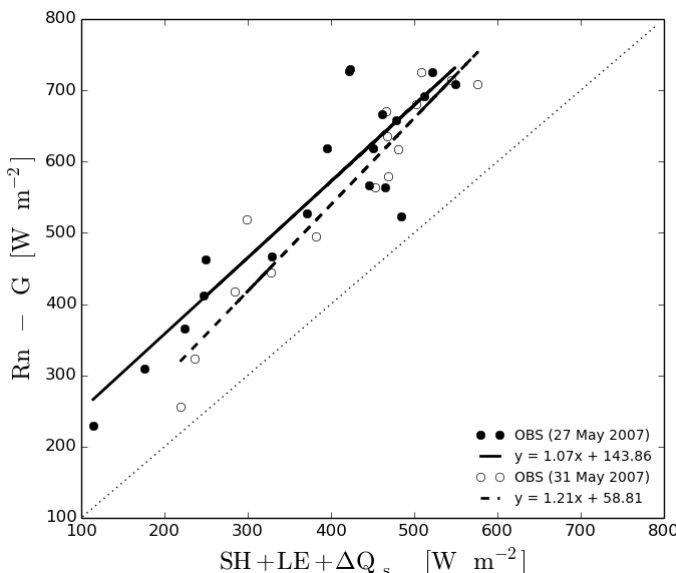

**Figure 3: Observed non-closure of the surface energy balance on 27 and 31 May 2007 during the CHATS experiment.**

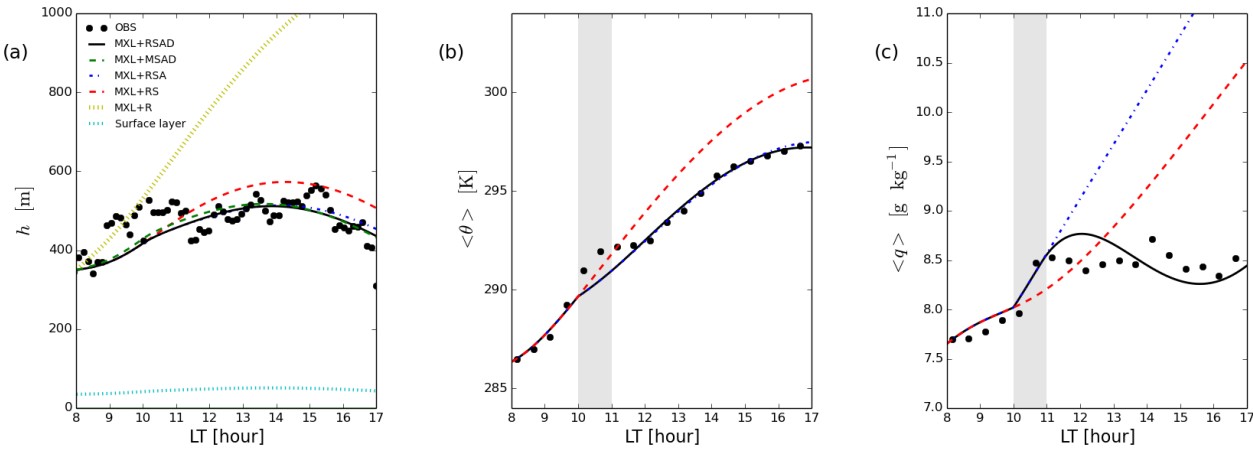

**Figure 4: Temporal evolution of the observed versus modelled mixed-layer quantities on 27 May 2007: (a) boundary layer height ($h$) (b) potential temperature, $\langle\theta\rangle$, and (c) specific humidity, $\langle q\rangle$. Observations are denoted by black symbols. $\langle\theta\rangle$ and $\langle q\rangle$ are measured at 29 m above the ground surface and $h$ is obtained from LIDAR data (Mayor 2011; Patton et al 2011). The numerical experiments are described in Table 1. Shaded areas in (b) and (c) indicate the**

10  **cooling and moistening periods of the atmospheric boundary layer.**





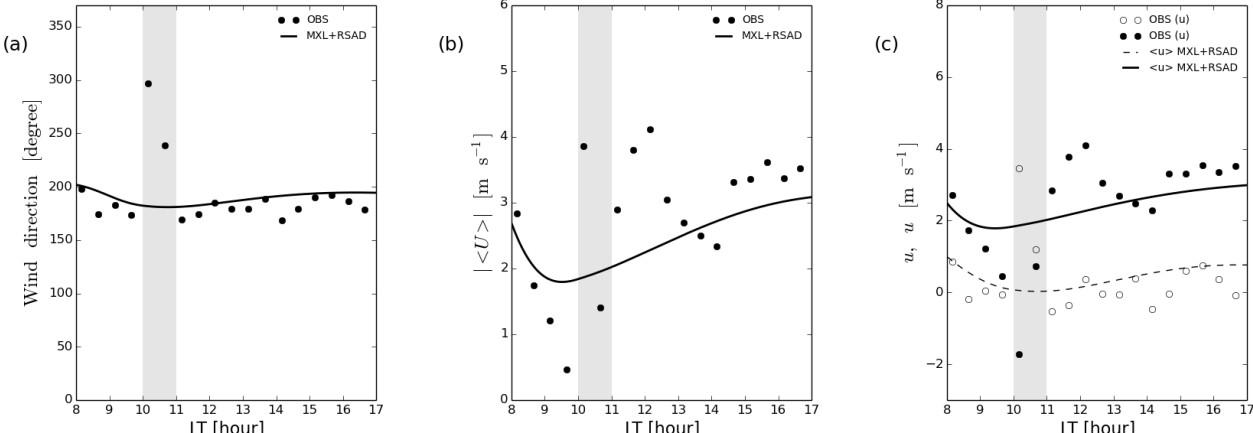

**Figure 5: Temporal evolution of the observed versus modelled boundary-layer dynamics at 29 m above the ground surface: (a) mixed-layer wind direction, (b) calculated modulus of the mixed-layer wind speed, (c) mixed-layer wind speed components. Shaded area indicates the period when the wind change occurs.**





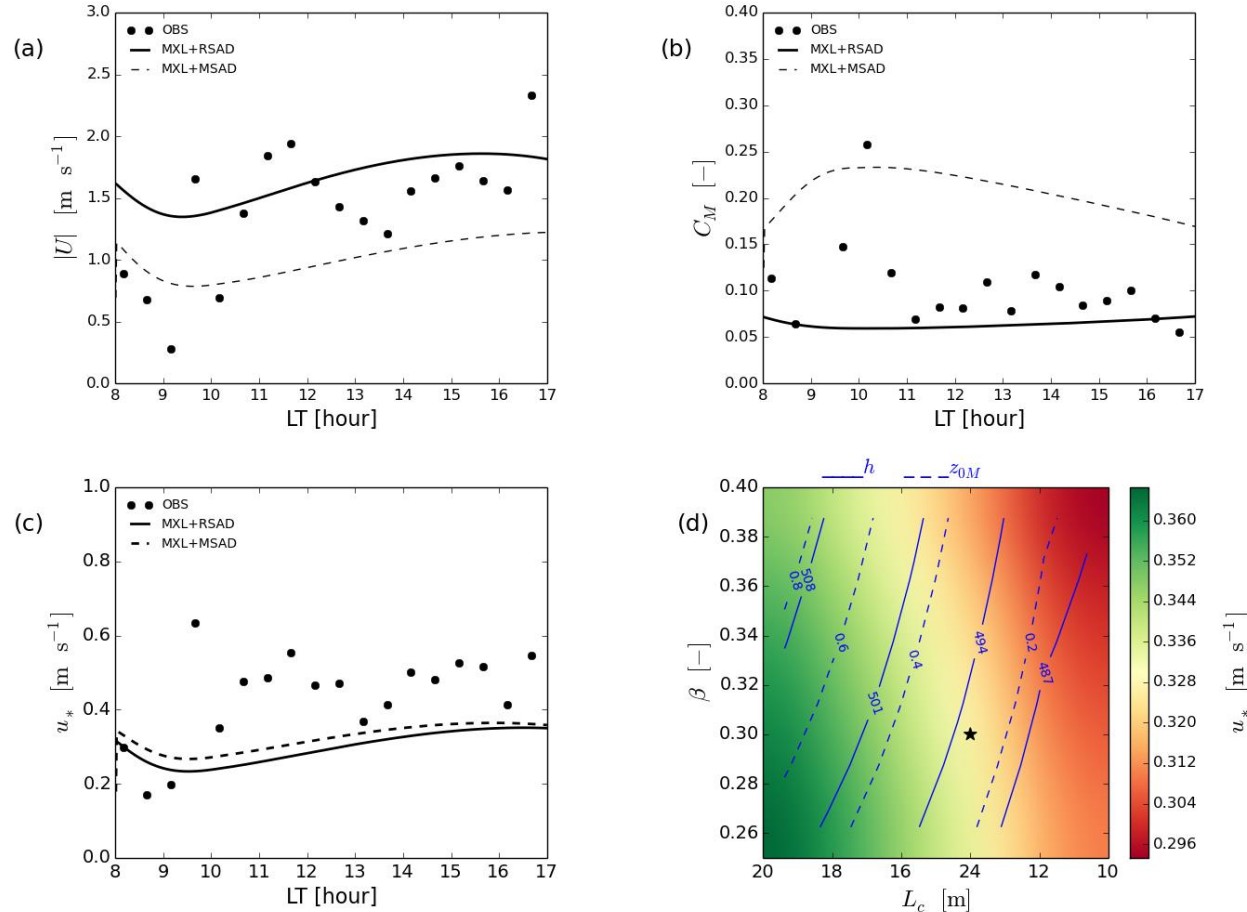

**Figure 6**: **Observed versus modelled modulus of the wind speed (a), momentum drag coefficient (b), friction velocity (c) with and without the RSL effects (solid line and dashed lines, respectively) at 10 m above the ground surface (equal to average tree heights, $h_c$ = 10 m). (d) Sensitivity of the friction velocity (colour scale), roughness length for momentum ($z_{0M}$ [m], dashed line) and boundary-layer height ($h$ [m], full line) at 13:00 LT to changes in the values of $\beta$ and $L_c$. The black asterisk indicates the conditions for the case study of 27 May 2007.**





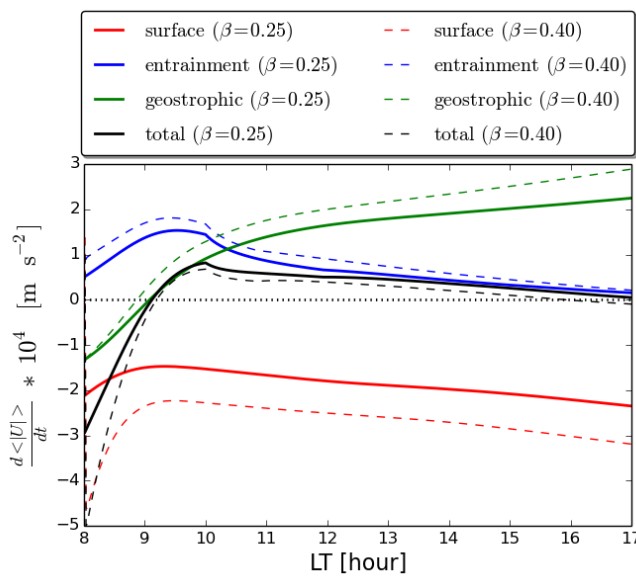

**Figure 7: Budget of the mixed-layer wind speed components $\langle|U|\rangle$ based on different canopy-flow forcing.**

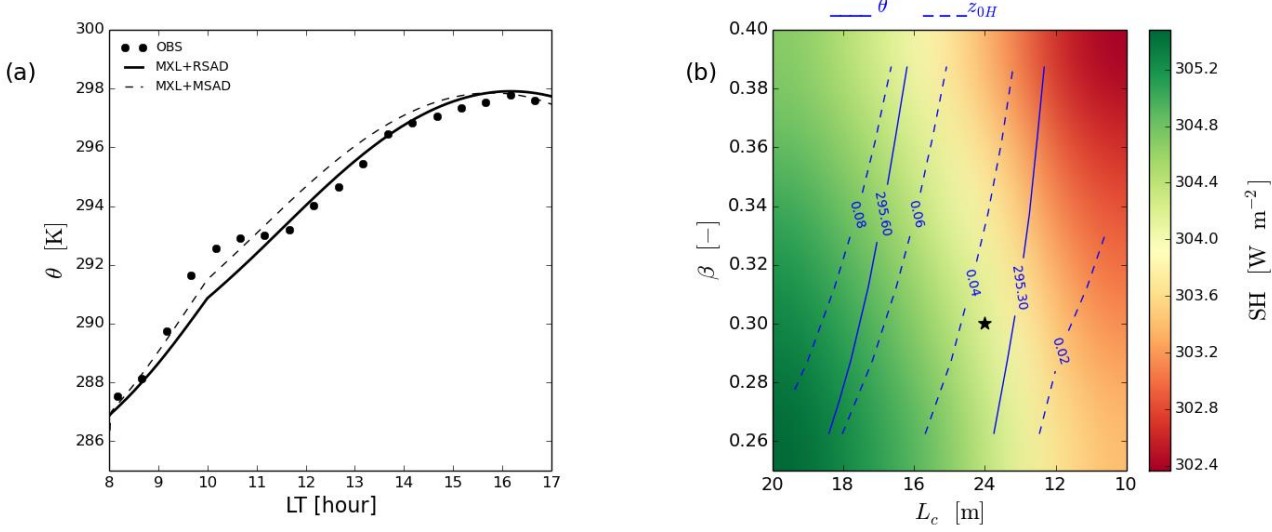

**Figure 8: (a) Temporal evolution of the observed versus modelled potential temperature, $\theta$, with and without the RSL effects at canopy-top level. (b) Effects of $\beta$ and $L_c$ on sensible heat flux (SH), $\theta$ and roughness length for heat ($z_{0H}$) at 13:00 LT. The black asterisk indicates the conditions and the results of the case study of 27 May 2007.**



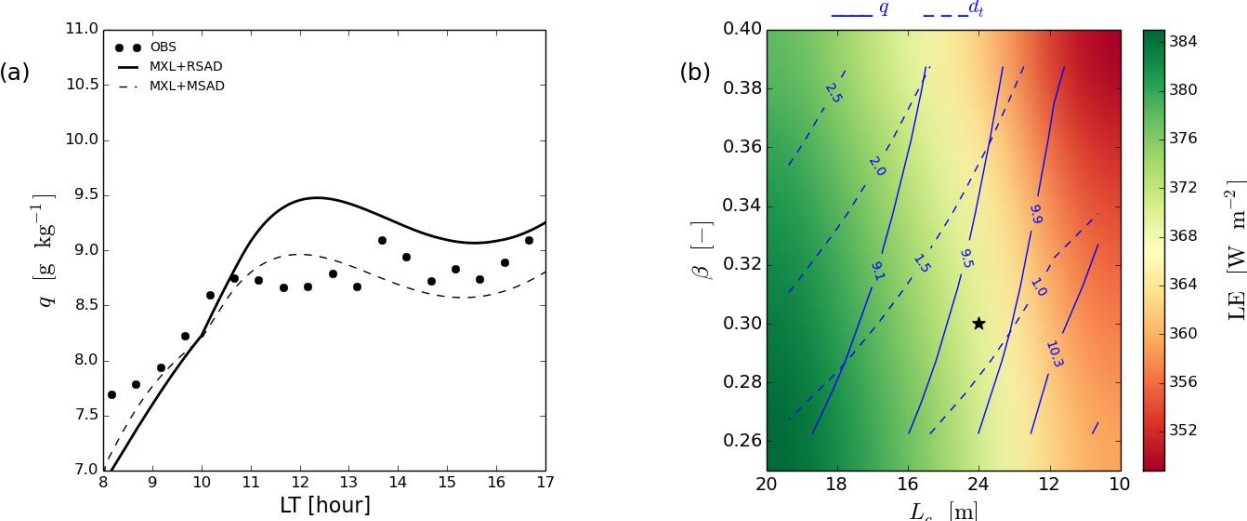

**Figure 9: (a) Temporal evolution of the observed versus modelled specific humidity, $q$, with and without the RSL effects at canopy-top level. (b) Effects of stability dependent $\beta$ and $L_c$ at canopy top on sensible heat flux (LE), $q$ and the effective displacement height ($d_t$) at 13:00 LT. The black asterisk indicates the conditions and the results of the case study (27 May 2007).**

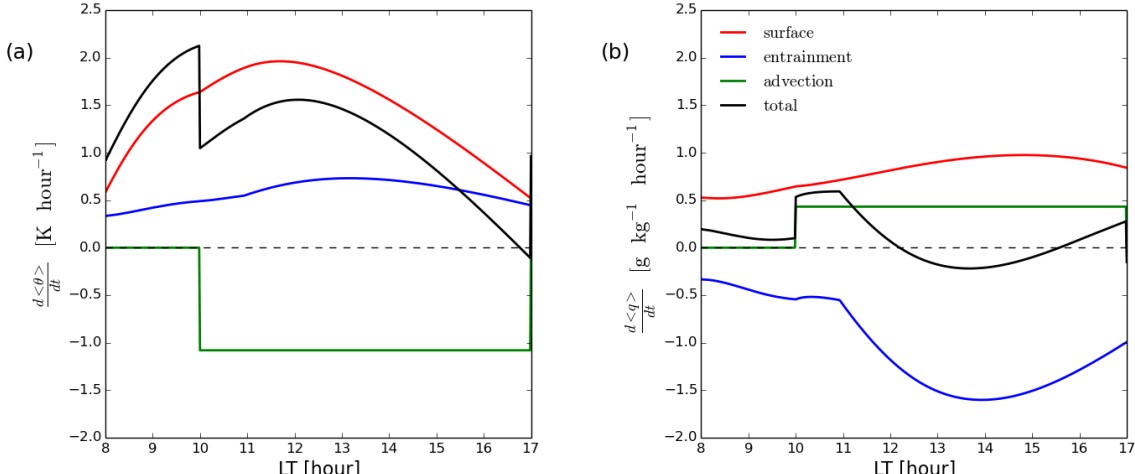

**Figure 10: Budget of the mixed-layer potential temperature (a) and specific humidity (b) calculated for the case study of 27 May 2007.**