# Peer review of "Integrating canopy and large-scale atmospheric effects in convective boundary-layer dynamics during CHATS experiment"

_Atmospheric Chemistry and Physics, 2016_

## Referee Comment (RC1) · Anonymous Referee #3 · 24 Oct 2016

This paper reports on an 'exploratory study of the potential alterations to the boundary-layer dynamics as calculated by large-scale models, when the roughness sublayer (RSL) is taken into account.' The authors conclude that (1) the RSL has a very limited effect on CBL dynamics (because the surface fluxes are affected only slightly), and that (2) when comparing simulated mean quantities and transfer coefficients near the canopy top with observations, it is important to account for the RSL.

This is a relevant and useful conclusion. I have several remarks, though:

\* A major shortcoming is that no quantitative error statistics are used to underpin statements of model performance. One has to judge model performance by looking at figures (eg Fig 8,9) to visually inspect the deviation of the model result (lines) versus

the observations (dots). It should be easy to add error statistics (RMSE, R2, bias, ...), and it will make the paper more rigorous.

* The authors take the 29-m level as representative for the mixed layer; I tend to disagree with this, so, unless the authors provide arguments for their claim, I would consider the 29-m as being much too low to represent the mixed layer.

* On the days considered in this study, CBL dynamics appears to be dominated by large-scale effects (advection, subsidence, ...). (See also p.10: "The analysis presented in Fig.4 shows that the complex boundary-layer structure at the CHATS site is highly dependent on the large-scale effects, including subsidence, advective cooling and moistening, as well as entrainment of dry air from the free troposphere.") Hence, I am wondering whether this case is the most appropriate for studying the impact of the RSL on the CBL.

* The authors say on p9l1-2 that "modelled SH & LE are likely to be the more correct values" (as compared to the observed values). I agree with that statement, but then I don't understand why they use data that are clearly not correct (i.e., the energy balance isn't closed) to validate their model. In fact, now you have a situation where the authors say, 'OK, the data aren't entirely correct, but we conclude that the model is performing fine anyway'. Hence I also question the statement "The comparison presented here confirms that our modelling system is capable of reproducing the diurnal variations in radiation and surface energy balance with sufficient accuracy" (p9l4-6).
* * *
minor remarks:

p1l28: "turbulent exchange of energy, momentum and matter between the Earth's surface and the free troposphere" - in this description you short-circuit the atmospheric boundary layer, perhaps better to replace 'free troposphere' by 'lower atmosphere'?

p2l29: I presume 'potential' ought to be 'potential temperature'

p3l20: It would be useful to include a figure (map) showing the measurement site and surroundings.

p4l14: sublayers => sublayers

p5Eq6: the slash in Eq 6 is not OK (should be slant and not vertical)

p5l20: 'heightd' => 'height d'

p6l8 and l11-12: 'strong unstable' => 'strongly unstable'

p7l3: what is 'toggled large-scale forcing'?

Fig.2: Observed G (soil heat flux) appears small (especially given the sparse canopy) - is this the value at the ground surface or at 5 cm depth? This could make a big difference, and explain the model-vs-observation discrepancy (and partly explain energy balance non-closure).

p10l14: 'on time' => 'with time' (?)

p.10: On page 10 you make a lot of assumptions: 'probably related to the sea breeze', 'probably related to drying associated with entrainment' etc..., using these to (try to) explain the simulated profiles' tendencies. All these 'probablies', are not very re-assuring and highly speculative. Maybe reconsider how you present all this in a more convincing way.

Table A1.1: Mentions 'lateral' wind speed component several times, shouldn't this be 'latitudinal' instead (to be consisten with the 'longitudinal' component)? Also: for the quantity CGsat in Table A1.1, the units seem odd, please check.

Fig 5(c) shows the u component of the wind speed twice, I guess the labelling should be changed to include both u and v

p15l6: "By applying the roughness sublayer formulations within the surface scheme of the model, the representation of the diurnal evolution of the boundary layer state

variables and the corresponding drag coefficients at the canopy height is improved." =>
this isn't so clear, e.g. in the case of specific humidity rather the contrary would appear
to be true (Fig.9a). Again, such statements should be underpinned by quantitative error
statistics (see remark above).

───────────────────────────

---

## Referee Comment (RC2) · Anonymous Referee #1 · 31 Oct 2016

This manuscript describes the inclusion of a model for the roughness sublayer into a column model. Results are compared to observations during the CHATS experiment, during which largescale effects on quantities of the ABL were of importance. Overall, the results of the manuscript appear to be valid and of interest to the readership of ACP. I therefore recommend publication of the manuscript pending the revisions and comments outlined below.

General comments:

There seems to be a systematic problem with the fluxes as compared to the EC method, which deserves some additional discussion (see specific comments) to strengthen the overall results of the paper.

Both days discussed in this manuscript have strong influence of largescale processes, which are difficult to quantify (and allow for adjusting of results to measurements). The paper would be greatly strengthened. In my opinion, while this shows that the model can be used for realistic conditions, the paper would be greatly strengthened by including an ideal day with no largescale forcing.

In general, some of the figures should be enhanced to improve legibility (fontsizes, and line thickness)

Specific comments:

P1 L26: The atmospheric boundary layer (ABL), as a part of the global climate, is a dynamic system that is highly dependent . . . –> The ABL may be part of the climate system, but is in my opinion not climate itself. Please rephrase

P2 L5: These structures are responsible for most of the momentum (70%) and turbulent kinetic energy (90 %) exchange between canopy and atmosphere Fininnigan, 2000; Finnigan et al., 2009) –> these numbers are in my opinion not generalizable, please substitute with a more general formulation (e.g. majority).

P2 L30: Extending these previous works, our study aimed to elucidate the ABL system for real conditions, taking the representation of the RSL into account. –> This sounds a bit clumsy

Introduction: since the work is about the effects of the RSL, it would be good to provide the reader with some estimate of the vertical extent of the RSL, in which MOST does not apply. This could be order of canopy heights or some scaling with respect to u*, LAI, hc

Figure 1: please make sure that all variables are explained in the caption. I find the use of hc for canopy height and h for MLH confusing. I assume that there is a temporal component in that Figure as the ABL grows from left to right. Please explain this as well in the caption. Also, it would be good if the text would mention before the Figure,

what are the variables that are actually predicted by the model

P6 L3: please provide equation for c_d, since this is the variable affecting L_c. Also, could you provide some information about the choice of a(z) = const. How much of a difference does this make?

P7 L21: We used the observations at the highest measurement level at the tower (29 m above ground surface) to evaluate the model results away from the canopy, where the RSL effects are minimal.–> Please justify and compare to likely RSL height. 29m is probably not representative of the MLH as a whole. I understand in the absence of profiles, compromises have to be made, but they should be articulated.

P8 L8: Figure 2a,b shows the observed and modelled components of the net radiation: downwelling (âĘŞ) and upwelling (âĘŚ) shortwave (SW)... –> This may be a good time to remind the reader how fluxes are modeled, as this if important to assess the difference between EC and model

Figure 3 and associated text: It is well know that EC leaves fluxes unclosed. However, I have two comments based on Figure 2 and 3. (1) Please switch the axes in Figure 3 as it is commonly done. (2) In forest canopies energy and moisture storage inside the canopy can play a role on the diurnal scale. So that EB closure should also be looked at as the daily integral of fluxes (unless storage is other wise accounted for). Also, Modeled fluxes seem to be systematically worse in the afternoon. Is there a reason for this?

P11 L19: Both C_M and |U|are altered in opposite directions, with magnitudes that fit the observation (Fig. 6a,b), thus leading to a relatively constant u* –> This behavior is not obvious to me from the methods section, please give some information about the mechanism and also please comment on the impact of the apparent difference between observed and modeled u*.

Figure 7 and associated text: Please provide some interpretation of the meaning of

this findings

P13 L22-25: In the absence of detailed observations of the temporal evolution at the entrainment zone, we are able to provide only first order estimates of the large scale effects relevant to our cases and discuss their impacts on the budgets of potential temperature and specific humidity (Fig. 10). –> See general comment about large-scale effects. In my opinion this is a limitation of the manuscript as these conditions can be used to make things work and warrants some discussion by the authors.

Figure 9b+10b: I find the sensitivity analysis for fluxes a bit confusing, given the fact that I don't know from the methods how these are related. If I understand the methods correctly, then the effect of beta and Lc on fluxes purely arises from changes in the displacement height. Or are there other effects at play.

P15 L9-11: However, due to compensation between the drag coefficients and the differences in the mean variables at two levels within the roughness sublayer, the modelled surface momentum and heat fluxes remain relatively unchanged (< 3 %). –> A similar argument probably applies to other fluxes. A critical reviewer might raise the question, what the advantage of the RSL formulation is, if it has little effect on the MLH and on fluxes (due to compensation of terms). I suggest that the authors add a sentence or two to explain why the RSL formulation matters based on the results presented.

Technical (not necessarily complete):

P7 L11: specific moments / a specific moment?

Figure 2: Please increase fontsize in figure

---

## Author Comment (AC1) · 10 Jan 2017

**First of all, we would like to thank Reviewer#3 for the valuable suggestions and comments. We have addressed all the comments raised by the referee in the response point by point and introduced the corresponding modifications in the manuscript. Below, we repeat the Reviewers' comments in normal font. Our replies are in bold-face and changes in the original manuscript are in italic.**

Overview:

This paper reports on an 'exploratory study of the potential alterations to the boundary-layer dynamics as calculated by large-scale models, when the roughness sublayer (RSL) is taken into account.' The authors conclude that (1) the RSL has a very limited effect on CBL dynamics (because the surface fluxes are affected only slightly), and that (2) when comparing simulated mean quantities and transfer coefficients near the canopy top with observations, it is important to account for the RSL. This is a relevant and useful conclusion. I have several remarks, though:

Specific Comments:

1) A major shortcoming is that no quantitative error statistics are used to underpin statements of model performance. One has to judge model performance by looking at figures (eg Fig 8,9) to visually inspect the deviation of the model result (lines) versus the observations (dots). It should be easy to add error statistics (RMSE, R2, bias, ...), and it will make the paper more rigorous.

**Answer: We agree with the referee's comment and suggestion. We performed the model vs. observations mean absolute error (MAE) statistics and placed the results in Table 2. We then refer to this error statistics in Table 2 when discussing the results in Figs. 6a, 8a, and 9a.**

*New:*
*Table 2: Calculated mean absolute error (MAE) of MXL+MSAD and MXL+RSAD numerical runs with respect to observations. The values of the MAE are presented in units of the corresponding quantities; the values in brackets show the model percentage of the MAE values relative to the daily means (between 08:00 and 17:00 LT) of the observed quantities respectively.*

| | $|U(z_r)|$ $[m\ s^{-1}]$ | $C_M(z_r)$ $[-]$ | $u_*$ $[m\ s^{-1}]$ | $\theta(z_r)$ $[K]$ | $q(z_r)$ $[g\ kg^{-1}]$ | $SH$ $[W\ m^{-2}]$ | $LE$ $[W\ m^{-2}]$ | $h$ $[m]$ |
|---|---|---|---|---|---|---|---|---|
| *Mean observed* | *1.45* | *0.11* | *0.44* | *293.86* | *8.60* | *128.46* | *250.88* | *473.06* |
| *MXL+MSAD* | | | | | | | | |
| *Mean model* | *1.00* | *0.20* | *0.32* | *294.37* | *8.49* | *222.53* | *313.72* | *463.84* |
| *MAE* | *0.50* | *0.10* | *0.13* | *0.47* | *0.22* | *87.18* | *59.22* | *34.09* |
| *(%)* | *(34.90)* | *(88.75)* | *(31.30)* | *(0.16)* | *(2.62)* | *(67.82)* | *(23.60)* | *(7.18)* |
| *MXL+RSAD* | | | | | | | | |
| *Mean model* | *1.64* | *0.06* | *0.30* | *294.01* | *8.81* | *217.38* | *307.26* | *457.80* |
| *MAE* | *0.34* | *0.06* | *0.15* | *0.41* | *0.37* | *81.81* | *52.86* | *35.97* |
| *(%)* | *(24.06)* | *(41.53)* | *(34.20)* | *(0.24)* | *(4.41)* | *(63.68)* | *(21.07)* | *(7.60)* |

*Table 2 shows the overview of the performance of the two numerical experiments with and without RSL representation (MXL+RSAD and MXL+MSAD, respectively) with respect to observations, as quantified by the mean absolute error (MAE). The numerical experiment with RSL representation performs better than the numerical experiment that omits the RSL when representing the wind speed and the drag at canopy height. Both numerical experiments (MXL+RSAD and MXL+MSAD) however underestimate the observed friction velocity. The small difference in magnitude of the friction velocity between the experiments is due to use of different roughness length and displacement height formulation: as stability dependent variables in MXL+RSAD, and as fixed parameters estimated under neutral condition in MXL+MSAD.   MXL+RSAD also*

*represents the potential temperature better than MXL+MSAD at the same level, but slightly overestimate the specific humidity. As expected, the largest MAEs are found for the surface fluxes (e.g. ~60 % MAE for SH with respect to the mean observed SH). Again, note that the observed SH and LE are not the 'true' surface fluxes since the energy balance is not closed (Fig. 3).*

2)  The authors take the 29-m level as representative for the mixed layer; I tend to disagree with this, so, unless the authors provide arguments for their claim, I would consider the 29-m as being much too low to represent the mixed layer.

**Answer: 29 m is the highest measurement level. We agree with the referee that this height is still in the surface layer. However, it is the closed to the mixed-layer characteristics. A deviation will indeed still be present, but since the surface layer is approximately 50 m at its deepest and the logarithmic profiles within the surface layer result in weaker deviations (with respect to mixed-layer values) in the upper part of that layer, the observations won't show strong deviations compared to mixed-layer values. This assumption is supported by the observations of the quantities of the two upper-most levels (23m and 29m). For instance, the slope derived from the potential temperature or specific humidity at 23 and 29m is less than 1% with respect to the vertical coordinate.**

**The following text is added in the manuscript to better explain the assumption of selecting the 29 m as a representative mixed-layer height in this study:**

*(New): The level of 29 m is considered to be representative of the mixed-layer values, since it is either located within the mixed layer or in the upper part of the surface layer, where deviations compared to mixed-layer values are small. Therefore, we employ it as the most representative of the mixed-layer characteristics.*

3)  On the days considered in this study, CBL dynamics appears to be dominated by large-scale effects (advection, subsidence, ...). (See also p.10: "The analysis presented in Fig.4 shows that the complex boundary-layer structure at the CHATS site is highly dependent on the large-scale effects, including subsidence, advective cooling and moistening, as well as entrainment of dry air from the free troposphere.") Hence, I am wondering whether this case is the most appropriate for studying the impact of the RSL on the CBL.

**Answer: As mentioned in the manuscript, in selecting the most appropriate days to carry out our research we define the following criteria: well-mixed boundary layer cloudless conditions, well-developed RSL (southerly winds during the entire day to maximize the effect of the footprint). In the entire period during the observations, mesoscale effects (e.g. horizontal fronts) were relevant, having a large impact on the diurnal variability of the measured quantities (Mayor 2011), similar as in our case studies (e.g. potential temperature drop of 1-2 K at around noon). These mesoscale effects have been previously studied and analyzed over the California Valley region where very active advection and topography driven flows where found (e.g. Zaremba; Carroll 1999; Bianco et al. 2011). We therefore took this opportunity to study the canopy effects on the CBL dynamics by also taking the large-scale effects into account in a systematic way.**

**Placed in more general context, there are several reasons why we chose the CHATS dataset as the main observational evidence to study the effects of RSL on the CBL-dynamics. High-quality measurements of the thermodynamics (and chemistry, used in our current work) is the first reason. Another reason is related to the canopy homogeneity in combination with the observed, relatively constant- wind direction, which allows a well-developed roughness sublayer above the canopy. This is convenient for studying canopy-atmosphere interaction in an 'idealized' way, since an irregular shape and distribution of the canopy would bring additional uncertainty in the turbulence structure within and above the canopy (Raupach et al. 1996; Finnigan et al. 2009).**

4)  The authors say on p9l1-2 that "modelled SH & LE are likely to be the more correct values" (as compared to the observed values). I agree with that statement, but then I don't understand why they use data that are clearly not correct (i.e., the energy balance isn't closed) to validate their model. In fact, now you have a situation where the authors say, 'OK, the data aren't entirely correct, but we conclude that the model is performing fine anyway'. Hence I also question the statement "The comparison presented here confirms that our modelling system is capable of reproducing the diurnal variations in radiation and surface energy balance with sufficient accuracy" (p9l4-6).

**Answer: Here we quote Foken (2008) with respect to energy balance closure: "The comparison of observational data and model output remains problematic". As discussed in Foken (2008), the reasons for the energy balance non-closer are related to the large scale turbulent structures, which the measurements in the surface layer are not able to capture. Due to this reasons, some studies even suggested that the energy balance (EB) closures should not be used as a quality criteria for turbulent fluxes (Aubinet et al. 1999). Nevertheless, we still use the sensible and the latent heat here, since we would like to compare the surface fluxes calculated with and without RSL parameterization, as shown in Table 2. We agree however with the referee's question about the statement "The comparison presented here confirms that our modelling system is capable of reproducing the diurnal variations in radiation and surface energy balance with sufficient accuracy". To make it more precise, we therefore have modified this statement into:**

*New: "The comparison presented here confirms that our modelling system is capable of reproducing the diurnal variations in radiation with sufficient accuracy. As in many other studies (see Foken 2008), the observed surface energy balance remains not closed, but with the deviations of similar magnitude as observed in other studies above high canopy.*

Minor remarks:

5)  p1l28: "turbulent exchange of energy, momentum and matter between the Earth's surface and the free troposphere" - in this description you short-circuit the atmospheric boundary layer, perhaps better to replace 'free troposphere' by 'lower atmosphere'?

**Answer: we agree with the referee's suggestion and replaced 'free troposphere' by 'lower atmosphere'. We consider this term more robust in the context of the statement.**

6)  p2l29: I presume 'potential' ought to be 'potential temperature'

**Answer: we corrected to 'potential temperature'.**

7)  p3l20: It would be useful to include a figure (map) showing the measurement site and surroundings

**Answer: we agree with the referee that it would be useful to include a figure (map) with the measurement site and surroundings. However, those figures and maps are already presented in the cited literature (Patton et al. 2011; Dupont; Patton 2012). Thus, in order not to overload the manuscript with figures, we have decided just to refer to the figures in these papers.**

8)  p4l14: sublayers => sublayer

**Answer: 'sublayers' corrected to 'sublayer'.**

9)  p5Eq6: the slash in Eq 6 is not OK (should be slant and not vertical)

**Answer: The referred vertical bar is one of the two vertical bars around $U(z_r)$ to denote that the modulus is used, similar to Eq. (3). For clarify a whitespace is inserted between the variables in Eq. (6).**

10) p5l20: 'heightd' => 'height d'

**Answer: 'heightd' has been corrected to 'height $d$'.**

11) p6l8 and l11-12: 'strong unstable' => 'strongly unstable'

**Answer: 'strong unstable' has been corrected to 'strongly unstable'**

12) p7l3: what is 'toggled large-scale forcing'?

**Answer: the 'toggled large scale forcing' refers to including or omitting subsidence, advection, free tropospheric drying at certain moment based on observations. We will delete this term however, since the sentence is clearer without it.**

13) Fig.2: Observed G (soil heat flux) appears small (especially given the sparse canopy)- is this the value at the ground surface or at 5 cm depth? This could make a big difference, and explain the model-vs-observation discrepancy (and partly explain energy balance non-closure).

**Answer: the soil heat flux ($G_m$) is measured at z = 5 cm depth. Then, the soil heat flux at the surface $G$ includes the heat storage in the soil, and is calculated as (Oliphant et al. 2004):**

$$G = G_m(z) + C_s \frac{\Delta T_s}{\Delta t} z,$$

**where $T_s$ is average soil temperature above the heat flux plate, $t$ is time and $C_s$ is soil heat capacity (see Oliphant et al. (2004) for details about the method for estimating $C_s$)**

**To be clearer, we have added the following sentence in the text:**

*New: "Note that presented G accounts for the heat storage in the soil, as calculated following Oliphant et al. (2004)."*

14) p10l14: 'on time' => 'with time' (?)

**Answer: 'on time' has been corrected to 'with time'**

15) p.10: On page 10 you make a lot of assumptions: 'probably related to the sea breeze', 'probably related to drying associated with entrainment' etc..., using these to (try to) explain the simulated profiles' tendencies. All these 'probablies', are not very re-assuring and highly speculative. Maybe reconsider how you present all this in a more convincing way.

**Answer: We have deleted the "probably "terms in our statements. We have also added relevant previous literature to support our hypothesis instead.**

*New: "We hypothesize that the rapid temperature drop before noon is related to the advection of cold air, due to a sea-breeze front, which is frequently observed around noon at the CHATS site (Mayor 2011)."*

*New: "After this increase, q remains steady until the end of the day (17:00 LT). We related this behavior of q after noon to the drying associated with the entrainment of free tropospheric (drier) air into the boundary layer, which can be driven by returned flow over the complex topography (Bianco et al. 2011). "*

16) Table A1.1: Mentions 'lateral' wind speed component several times, shouldn't this be 'latitudinal' instead (to be consisten with the 'longitudinal' component)? Also: for the quantity CGsat in Table A1.1, the units seem odd, please check.

**Answer: although the coordinate are presented in latitude and longitude, the term "lateral" is often used in the literature to define winds "from the side". We therefore prefer to use this term. As for the second part of the comment, we thank the reviewer for this specific comment about the units of the quantity of the saturated soil conductivity of heat is in units [J m$^{-3}$ K$^{-1}$], as stated in the table. This variable, modified for the soil moisture content, is multiplied by the soil heat flux to yield the soil temperature tendency.**

17) Fig 5(c) shows the u component of the wind speed twice, I guess the labelling should be changed to include both u and v

**Answer: The referee is correct. We have made new figure and corrected the typo.**

18) p15l6: "By applying the roughness sublayer formulations within the surface scheme of the model, the representation of the diurnal evolution of the boundary layer state variables and the corresponding drag coefficients at the canopy height is improved." => this isn't so clear, e.g. in the case of specific humidity rather the contrary would appear to be true (Fig.9a). Again, such statements should be underpinned by quantitative error statistics (see remark above).

**Answer: We agree with the referee that we should be more precise in our statements. In that respect, we modify the statement:**

*New: "In our modelling framework, and in general in the coupled land-atmosphere models, the representation of the surface fluxes is locked and controlled by the boundary conditions. The sensible and latent heat fluxes are bounded by the surface available energy, and the momentum flux is constrained by the pressure gradient and the entrainment of momentum, the latter dependent on the boundary-layer growth. In consequence, adding a roughness-sublayer representation in the surface scheme of the model, alters the partitioning of the surface fluxes (e.g. sensible and latent heat) through the altered roughness length and displacement height. Specifically for our case studies, the canopy's impact on convective boundary-layer dynamics is relatively minor, due to its small effect on modelled surface fluxes and the bulk boundary-layer properties well above the canopy ($z > 2h_c$). The tall canopy however strongly affects the mean gradients and transfer coefficients within the roughness sublayer. Thus, considering the roughness sublayer parameterization is important when comparing observations and large-scale model outputs of the mean quantities near and just above the canopy."*

**References**

Aubinet, M., and Coauthors, 1999: Estimates of the annual net carbon and water exchange of forests: the EUROFLUX methodology. *Advances in ecological research*, **30,** 113-175.

Bianco, L., I. V. Djalalova, C. W. King, and J. M. Wilczak, 2011: Diurnal Evolution and Annual Variability of Boundary-Layer Height and Its Correlation to Other Meteorological Variables in California's Central Valley. *Bound-Lay Meteorol*, **140,** 491-511.

Dupont, S., and E. G. Patton, 2012: Influence of stability and seasonal canopy changes on micrometeorology within and above an orchard canopy: The CHATS experiment. *Agric For Meteorol*, **157,** 11-29.

Finnigan, J. J., R. H. Shaw, and E. G. Patton, 2009: Turbulence structure above a vegetation canopy. *J Fluid Mech*, **637,** 387-424.

Foken, T., 2008: *Micrometeorology, 308 pp.* Springer Science & Business Media.

Mayor, S. D., 2011: Observations of Seven Atmospheric Density Current Fronts in Dixon, California*. *Mon Wea Rev*, **139,** 1338-1351.

Oliphant, A. J., and Coauthors, 2004: Heat storage and energy balance fluxes for a temperate deciduous forest. *Agric For Meteorol*, **126,** 185-201.

Patton, E. G., and Coauthors, 2011: The Canopy Horizontal Array Turbulence Study. *Bull Amer Meteor Soc*, **92,** 593-611.

Raupach, M. R., J. J. Finnigan, and Y. Brunet, 1996: Coherent eddies and turbulence in vegetation canopies: the mixing-layer analogy. *Bound-Lay Meteorol*, **78,** 351-382.

Zaremba, L. L., and J. J. Carroll, 1999: Summer Wind Flow Regimes over the Sacramento Valley. *J Appl Meteorol*, **38,** 1463-1473.

---

## Author Comment (AC2) · 10 Jan 2017

We thank Reviewer 1 for the constructive comments. We have addressed all the comments raised by the referee in the response point by point and introduced the corresponding modifications in the manuscript. Below, we repeat the Reviewers' comments in normal font. Our replies are in bold-face and changes in the original manuscript are in italic.

Overview:

This manuscript describes the inclusion of a model for the roughness sublayer into a column model. Results are compared to observations during the CHATS experiment, during which large-scale effects on quantities of the ABL were of importance. Overall, the results of the manuscript appear to be valid and of interest to the readership of ACP. I therefore recommend publication of the manuscript pending the revisions and comments outlined below.

General Comments:

1) There seems to be a systematic problem with the fluxes as compared to the EC method, which deserves some additional discussion (see specific comments) to strengthen the overall results of the paper.

**Answer: Here we quote Foken (2008) with respect to energy balance closure: "The comparison of observational data and model output remains problematic". As discussed in Foken (2008), the reasons for the energy balance non-closer are related to the large scale turbulent structures, which the measurements in the surface layer are not able to capture. Due to this reasons, some studies even suggested that the energy balance (EB) closures should not be used as a quality criteria for turbulent fluxes (Aubinet et al. 1999). Nevertheless, we still use the sensible and the latent heat here, since we would like to compare the surface fluxes calculated with and without RSL parameterization. To make it more precise, we have modified the concluding statement (P9 L4-6): "The comparison presented here confirms that our modelling system is capable of reproducing the diurnal variations in radiation and surface energy balance with sufficient accuracy". The new statement reads:**

*New: "The comparison presented here confirms that our modelling system is capable of reproducing the diurnal variations in radiation with sufficient accuracy. As in many other studies (see Foken 2008), the observed surface energy balance remains not closed, but with the deviations of similar magnitude as observed in other studies above high canopy.*

2) Both days discussed in this manuscript have strong influence of largescale processes, which are difficult to quantify (and allow for adjusting of results to measurements). In my opinion, while this shows that the model can be used for realistic conditions, the paper would be greatly strengthened by including an ideal day with no large scale forcing.

**Answer: As mentioned in the manuscript, in selecting the most appropriate days to carry out our research we define the following criteria: well-mixed boundary layer cloudless conditions, well-developed RSL (southerly winds during the entire day to maximize the effect of the footprint). In the entire period during the observations, mesoscale effects (e.g. horizontal fronts) were relevant, having a large impact on the diurnal variability of the measured quantities (Mayor 2011), similar as in our case studies (e.g. potential temperature drop of 1-2 K at around noon). These mesoscale effects have been previously studied and analyzed over the California Valley region where very active advection and topography driven flows where found (e.g. Zaremba; Carroll 1999; Bianco et al. 2011). We therefore took this opportunity to study the canopy effects on the CBL dynamics by also taking the large-scale effects into account in a systematic way.**

**Placed in more general context, there are several reasons why we chose the CHATS dataset as the main observational evidence to study the effects of RSL on the CBL-dynamics. High-quality measurements of the thermodynamics (and chemistry, used in our current work) is the first reason. Another reason is**

related to the canopy homogeneity in combination with the observed, relatively constant- wind direction, which allows a well-developed roughness sublayer above the canopy. This is convenient for studying canopy-atmosphere interaction in an 'idealized' way, since an irregular shape and distribution of the canopy would bring additional uncertainty in the turbulence structure within and above the canopy (Raupach et al. 1996; Finnigan et al. 2009).

3) In general, some of the figures should be enhanced to improve legibility (font sizes, and line thickness).

**Answer: We find this remark of the referee to improve the visualization in several figures. Thus, we increased the font size and readability in Fig. 2,3,4,5.**

Specific comments:

4) P1 L26: The atmospheric boundary layer (ABL), as a part of the global climate, is a dynamic system that is highly dependent . . . –> The ABL may be part of the climate system, but is in my opinion not climate itself. Please rephrase.

**Answer: We modified the statement as follows:**

*New: "The atmospheric boundary layer (ABL), as a component of the global climate system, is characterized by the turbulent exchange of energy, momentum and matter between the Earth's surface and the lower atmosphere, as well as by the influence of larger-scale atmospheric processes (Stull 1988)."*

5) P2 L5: These structures are responsible for most of the momentum (70%) and turbulent kinetic energy (90 %) exchange between canopy and atmosphere Fininnigan, 2000; Finnigan et al., 2009) –> these numbers are in my opinion not generalizable, please substitute with a more general formulation (e.g. majority).

**Answer: We agree with the referee's remark; since the statement belongs to the introduction, we can be more general. We used the following modification:**

*New: "These structures are responsible for majority of the momentum and turbulent kinetic energy exchange between canopy and atmosphere (Finnigan 2000; Finnigan et al. 2009)."*

6) P2 L30: Extending these previous works, our study aimed to elucidate the ABL system for real conditions, taking the representation of the RSL into account. –> This sounds a bit clumsy

**Answer: we slightly modified this statement:**

*New: "Here, we extend on previous studies by analyzing the impact of the RSL representation on the dynamic evolution of the ABL constrained and evaluated with available observations."*

7) Introduction: since the work is about the effects of the RSL, it would be good to provide the reader with some estimate of the vertical extent of the RSL, in which MOST does not apply. This could be order of canopy heights or some scaling with respect to u*, LAI, hc.

**Answer: We modify the following sentence in P2L5 to inform the reader about the vertical extent of the RSL:**

*New: "These structures are responsible for majority of the momentum and turbulent kinetic energy exchange between canopy and atmosphere (Finnigan, 2000; Finnigan et al., 2009). Dependent on canopy density and height, as well as atmospheric diabatic stability, the vertical extent of the RSL is estimated to reach up to 2-3 canopy heights (Dupont; Patton 2012; Shapkalijevski et al. 2016),."*

8) Figure 1: please make sure that all variables are explained in the caption. I find the use of hc for canopy

height and h for MLH confusing. I assume that there is a temporal component in that Figure as the ABL grows from left to right. Please explain this as well in the caption. Also, it would be good if the text would mention before the Figure, what are the variables that are actually predicted by the model?

**Answer: We modified the text in the caption of Fig. 1 to better explain the figure and introduce all the variables, including the ones calculated by the MXLCH model:**

**Old: "Figure 1: Schematic overview of the coupled land-vegetation-atmospheric mixed-layer model, with both including and omitting the RSL effects in the flux-profile relationships. The vertical origin of the co-ordinate system is placed at the displacement height _d_. The height of the surface layer is calculate as 10 % of the boundary-layer height (Stull 1988)."**

*New: "Figure 1: Schematic overview of the coupled land-vegetation-atmospheric system and its representation in the mixed-layer model. The vertical origin of the co-ordinate system is placed at the displacement height d. The height of the surface layer is estimated as 10 % of the boundary-layer height (Stull 1988). The scheme illustrates the diurnal (convective) evolution of the boundary-layer height (h) and stability dependent roughness lengths for momentum and scalars ($z_{0M}$ and $z_{0H}$). Profiles of boundary-layer state variables (wind speed, |U|, potential temperature, $\langle\theta\rangle$, and specific humidity, $\langle q\rangle$), are also presented, both including and omitting the RSL effects in the flux-gradient relationships."*

**For the second part of the referee's comment about the predicted variables by the model, we placed a note (sentence) in the text before Fig. 1 the the model variables are explained later in this section.**

9) P6 L3: please provide equation for $c_d$, since this is the variable affecting $L_c$. Also, could you provide some information about the choice of a(z) = const. How much of a difference does this make?

**Answer: In our modelling framework, $c_d = \left(\frac{u_*}{|U|}\right)^2$, and is calculated at the canopy top. We included this equation in the text (P6 L3) as follows:**

*New: " .., while $c_d$ is the leaf drag coefficient, calculated from the observations at the canopy top ($c_d = u_*^2/|U|^2$). "*

**Next to that, the assumption that _a(z)_ is constant originates from Harman and Finnigan (2007), who assumed this for dense canopy. Shapkalijevski et al., (2016) showed that this assumption holds for the fully vegetated CHATS canopy. Finally, apart from this study, but related to the referee's question, Ouwersloot et al., (2016) by using high-resolution large eddy simulation over canopy under neutral conditions found that the impact of applying _a_ either constant or non-constant in height _a_ has small impact on the profiles of wind speed and shear within and above the canopy.**

10) P7 L21: We used the observations at the highest measurement level at the tower (29 m above ground surface) to evaluate the model results away from the canopy, where the RSL effects are minimal.–> Please justify and compare to likely RSL height. 29m is probably not representative of the MLH as a whole. I understand in the absence of profiles, compromises have to be made, but they should be articulated.

**Answer: 29 m is the highest measurement level. We agree with the referee that this height is still in the surface layer. However, it is the closed to the mixed-layer characteristics. A deviation will indeed still be present, but since the surface layer is approximately 50 m at its deepest and the logarithmic profiles within the surface layer result in weaker deviations (with respect to mixed-layer values) in the upper part of that layer, the observations won't show strong deviations compared to mixed-layer values. This assumption is supported by the observations of the quantities of the two upper-most levels (23m and 29m). For instance, the slope derived from the potential temperature or specific humidity at 23 and 29m is less than 1% with respect to the vertical coordinate.**

**The following text is added in the manuscript to better explain the assumption of selecting the 29 m as a representative mixed-layer height in this study:**

*(New): The level of 29 m is considered to be representative of the mixed-layer values, since it is either located within the mixed layer or in the upper part of the surface layer, where deviations compared to mixed-layer values are small. Therefore, we employ it as the most representative of the mixed-layer characteristics.*

11) P8 L8: Figure 2a,b shows the observed and modelled components of the net radiation: downwelling and upwelling shortwave (SW) ´ . . . –> This may be a good time to remind the reader how fluxes are modeled, as this if important to assess the difference between EC and model.

**Answer: Since the procedure of modelling the radiation and surface fluxes is already demonstrated and evaluated in a number of studies (e.g. van Heerwaarden et al. 2009; Ouwersloot et al. 2012; van Stratum et al. 2012; Vilà-Guerau de Arellano et al. 2015), here (ate the place suggested by the referee) we have placed a general explanation to inform the reader:**

*New: "the surface fluxes in the model are calculated from the differences between the surface and the roughness sublayer (reference height) values of the mean quantities and the transfer coefficients for momentum and scalars."*

12) Figure 3 and associated text: It is well know that EC leaves fluxes unclosed. However, I have two comments based on Figure 2 and 3. (1) Please switch the axes in Figure 3 as it is commonly done; (2) In forest canopies energy and moisture storage inside the canopy can play a role on the diurnal scale. So that EB closure should also be looked at as the daily integral of fluxes (unless storage is otherwise accounted for). Also, Modeled fluxes seem to be systematically worse in the afternoon. Is there a reason for this?

**Answer:**

**(1) We switched the axes in Fig. 3.**
**(2) Regarding the energy and moisture storage inside the canopy, we refer to P8 L20-27 in the manuscript, where we expressed that and how we included the storage terms in the energy balance.**

**Finally, instead of the referee' s statement "Modeled fluxes seem to be systematically worse in the afternoon", we conclude that modelled fluxes deviate systematically ,ore from EC fluxes in the afternoon. In spite of the difficulty in reproducing local process driven by the canopy at the surface and the large-scale effects at around noon, modelled surface fluxes were systematically worst in the early afternoon, when the effects of the large-convective (boundary-layer) eddies on the surface turbulence are expected to be larger (Zilitinkevich et al. 2006).**

13) P11 L19: Both $C_M$ and $|U|$ are altered in opposite directions, with magnitudes that fit the observation (Fig. 6a,b), thus leading to a relatively constant u* –> This behavior is not obvious to me from the methods section, please give some information about the mechanism and also please comment on the impact of the apparent difference between observed and modeled u*.

**Answer: To make the statement clearer, first we have corrected and modified it. The modified sentence is:**

*New: "Both $C_M$ and $|U|$ are altered in opposite directions when the RSL representation is introduced (Eq. 4 and 5), with magnitudes that fit the observation (Fig. 6a,b), thus leading to a relatively unchanged $u_*$ (see Eq. 6)".*

**Second, we noticed a mistake in the published formulation of the friction velocity (Eq. 6):**

$$u_* = \sqrt{C_M(z_r)|U(z_r)|},$$

**where the wind speed modulus should be outside the "√" operator. Thus the modified and corrected formulation is:**

*New:*
$$u_* = \sqrt{C_M(z_r)}\,|U(z_r)|. \tag{6}$$

**The main mechanism for the similarity in $u_*$ of the model runs with and without the roughness sublayer effects, as discussed in the manuscript on page 11 and line 08 – 14, involves canopy effects on the drag $C_M$ and $|U|$. $C_M$ is decreased 4 order of magnitudes, while $|U|$ is increased by 50% when RSL is included. Consequently, the resulting $u_*$ remains relatively unchanged (Eq. 6). Physically, this can be explained by the presence of an inflection point of the mean wind speed at canopy vicinity, which leads to smaller drag and thus larger wind speed (but smaller gradients) within the RSL than postulated by the standard similarity theory.**

**Finally, the underestimation of the observed $u_*$ for both numerical experiments is commented on page 11 lline 16:**

**"Both the MXL+RSAD and MXL+MSAD model runs, i.e., with and without the effects of the RSL included, underestimate $u_*$ by about 30% with respect to the observed daily average (Fig. 6c and Table 2).**

14) Figure 7 and associated text: Please provide some interpretation of the meaning of this findings.

**Answer: On page 12 line 9 we added the following concluding sentence:**

*New: In summary, although the variation of the RSL scale β strongly affects the surface shear partitioning in the momentum budget, the total momentum tendency remains relatively unchanged due to compensation by the geostrophic and entrainment contribution. This means that the imposed pressure gradient force, integrated over the BL-depth is balanced by the surface friction and momentum entrainment. Since the boundary-layer depth is similar between the both runs, then pressure gradient force and momentum entrainment are altered to balance the differences in the surface shear between the runs.*

15) P13 L22-25: In the absence of detailed observations of the temporal evolution at the entrainment zone, we are able to provide only first order estimates of the large scale effects relevant to our cases and discuss their impacts on the budgets of potential temperature and specific humidity (Fig. 10). –> See general comment about largescale effects. In my opinion this is a limitation of the manuscript as these conditions can be used to make things work and warrants some discussion by the authors.

**Answer: We are aware of this limitation in our study. Please see the answer to general comment (2) for more explanation. Here we would like to state that although the large scale forcing strongly affects the CBL dynamics over CHATS (as mentioned several times in the manuscript), they will equally affect both numerical runs with and without RSL representation. Thus, we can conclude that the results in this study about the RSL effects on the CBL dynamics are still relevant when considering the large-scale processes. The contribution of the RSL effects on the budgets of the (thermos-)dynamic quantities, compared to the contribution by the large scale processes is much smaller however.**

16) Figure 9b+10b: I find the sensitivity analysis for fluxes a bit confusing, given the fact that I don't know from the methods how these are related. If I understand the methods correctly, then the effect of beta and Lc on fluxes purely arises from changes in the displacement height. Or are there other effects at play.

**Answer: Performed sensitivity analysis showed that the modeled surface fluxes are affected by the variation of RSL scales (atmospheric stability dependent $\beta$ and $L_c$) via the changes in the displacement**

**height and the stability dependent roughness lengths for momentum and scalars. There are no other effects in play. This is stated on page 13 lines 4-9.**

17) P15 L9-11: However, due to compensation between the drag coefficients and the differences in the mean variables at two levels within the roughness sublayer, the modelled surface momentum and heat fluxes remain relatively unchanged (< 3 %). –> A similar argument probably applies to other fluxes. A critical reviewer might raise the question, what the advantage of the RSL formulation is, if it has little effect on the MLH and on fluxes (due to compensation of terms). I suggest that the authors add a sentence or two to explain why the RSL formulation matters based on the results presented.

**Answer: We have modified the last paragraph of the Conclusions section to better explain why the RSL matters based on presented results:**

*New: "In our modelling framework, and in general in the coupled land-atmosphere models, the representation of the surface fluxes is locked and controlled by the boundary conditions. The sensible and latent heat fluxes are bounded by the surface available energy, and the momentum flux is constrained by the pressure gradient and the entrainment of momentum, the latter dependent on the boundary-layer growth. In consequence, adding a roughness-sublayer representation in the surface scheme of the model, alters the partitioning of the surface fluxes (e.g. sensible and latent heat) through the altered roughness length and displacement height. Specifically for our case studies, the canopy's impact on convective boundary-layer dynamics is relatively minor, due to its small effect on modelled surface fluxes and the bulk boundary-layer properties well above the canopy ($z > 2h_c$). The tall canopy however strongly affects the mean gradients and transfer coefficients within the roughness sublayer. Thus, considering the roughness sublayer parameterization is important when comparing observations and large-scale model outputs of the mean quantities near and just above the canopy."*

Technical (not necessarily complete):

18) P7 L11: specific moments / a specific moment?

**Answer: we corrected as "..a specific moment.."**

19) Figure 2: Please increase font size in figure

**Answer: We increased the font size in Fig. 2.**

**References**

Aubinet, M., and Coauthors, 1999: Estimates of the annual net carbon and water exchange of forests: the EUROFLUX methodology. *Advances in ecological research*, **30,** 113-175.

Bianco, L., I. V. Djalalova, C. W. King, and J. M. Wilczak, 2011: Diurnal Evolution and Annual Variability of Boundary-Layer Height and Its Correlation to Other Meteorological Variables in California's Central Valley. *Bound-Lay Meteorol*, **140,** 491-511.

Dupont, S., and E. G. Patton, 2012: Momentum and scalar transport within a vegetation canopy following atmospheric stability and seasonal canopy changes: The CHATS experiment. *Atmos Chem Phys*, **12,** 5913-5935.

Finnigan, J., 2000: Turbulence in plant canopies. *Annu Rev Fluid Mech*, **32,** 519-571.

Finnigan, J. J., R. H. Shaw, and E. G. Patton, 2009: Turbulence structure above a vegetation canopy. *J Fluid Mech*, **637,** 387-424.

Foken, T., 2008: *Micrometeorology, 308 pp.* Springer Science & Business Media.

Mayor, S. D., 2011: Observations of Seven Atmospheric Density Current Fronts in Dixon, California*. *Mon Wea Rev*, **139,** 1338-1351.

Ouwersloot, H., A. Moene, J. Attema, and J. V.-G. De Arellano, 2016: Large-Eddy Simulation Comparison of Neutral Flow Over a Canopy: Sensitivities to Physical and Numerical Conditions, and Similarity to Other Representations. *Boundary-Layer Meteorology*, 1-19.

Ouwersloot, H. G., and Coauthors, 2012: Characterization of a boreal convective boundary layer and its impact on atmospheric chemistry during HUMPPA-COPEC-2010. *Atmos Chem Phys*, **12,** 9335-9353.

Raupach, M. R., J. J. Finnigan, and Y. Brunet, 1996: Coherent eddies and turbulence in vegetation canopies: the mixing-layer analogy. *Bound-Lay Meteorol*, **78,** 351-382.

Shapkalijevski, M., A. F. Moene, H. G. Ouwersloot, E. G. Patton, and J. V.-G. d. Arellano, 2016: Influence of Canopy Seasonal Changes on Turbulence Parameterization within the Roughness Sublayer over an Orchard Canopy. *J Appl Meteorol Clim*, **55,** 1391-1407.

Stull, R. B., 1988: *An introduction to boundary layer meteorology.* Vol. 13, Springer Science & Business Media, 670 pp.

van Heerwaarden, C. C., J. Vilà-Guerau de Arellano, A. F. Moene, and A. A. M. Holtslag, 2009: Interactions between dry-air entrainment, surface evaporation and convective boundary-layer development. *Q J Roy Meteorol Soc*, **135,** 1277-1291.

van Stratum, B. J. H., and Coauthors, 2012: Case study of the diurnal variability of chemically active species with respect to boundary layer dynamics during DOMINO. *Atmos. Chem. Phys.*, **12,** 5329-5341.

Vilà-Guerau de Arellano, J., C. C. van Heerwaarden, B. J. van Stratum, and K. van den Dries, 2015: *Atmospheric boundary layer: Integrating air chemistry and land interactions.* Cambridge University Press, 276 pp pp.

Zaremba, L. L., and J. J. Carroll, 1999: Summer Wind Flow Regimes over the Sacramento Valley. *J Appl Meteorol*, **38,** 1463-1473.

Zilitinkevich, S. S., and Coauthors, 2006: The influence of large convective eddies on the surface-layer turbulence. *Q. J. Roy. Meteor. Soc.*, **132,** 1426-1456.